# On permutation symmetries in Bayesian neural network posteriors: a variational perspective

**Simone Rossi**
Stellantis, France

**Ankit Singh**
Stellantis, India

**Thomas Hannagan**
Stellantis, France

## Abstract

The elusive nature of gradient-based optimization in neural networks is tied to their loss landscape geometry, which is poorly understood. However recent work has brought solid evidence that there is essentially no loss barrier between the local solutions of gradient descent, once accounting for weight-permutations that leave the network's computation unchanged. This raises questions for approximate inference in Bayesian neural networks (BNNs), where we are interested in marginalizing over multiple points in the loss landscape. In this work, we first extend the formalism of marginalized loss barrier and solution interpolation to BNNs, before proposing a matching algorithm to search for linearly connected solutions. This is achieved by aligning the distributions of two independent approximate Bayesian solutions with respect to permutation matrices. We build on the results of Ainsworth et al. (2023), reframing the problem as a combinatorial optimization one, using an approximation to the sum of bilinear assignment problem. We then experiment on a variety of architectures and datasets, finding nearly zero marginalized loss barriers for linearly connected solutions.

## 1 Introduction

Throughout the last decade, deep neural networks (DNNs) have achieved significant success in a wide range of practical applications, becoming the fundamental ingredient for e.g., computer vision [e.g., 57, 25, 41, 64], language models [e.g., 24, 82, 17] and generative models [e.g., 54, 96, 97, 98, 102, 34]. Despite recent important advancements, understanding the loss landscape of DNNs is still challenging. The characterization of its highly non-convex nature, its relation with architectural choices like depth and width and the connection with optimization and generalization are just some of the problems which have been the focus of extensive research in the last few years [e.g., 76, 26, 36, 32, 2, 30, 38, 79]. It is well known, for example, that one of the fundamental characteristics of deep neural networks is their ability to learn hierarchical features, and in this regards deeper networks seem to be exponentially more expressive than shallower models [e.g., 3, 5, 7, 8, 18, 109], leading the loss landscape to have many optima due to symmetries and over-parameterization [76, 26, 111, 94]. At the same time, the role of the depth of a model in relation with its width is far less understood [80], despite wide neural networks exhibiting important theoretical properties in their infinite limit behavior [e.g., 73, 23, 29, 53, 48, 37, 20].

Two notions that have been useful to shed light on the geometry of loss landscapes are that of *loss barriers* and *mode connectivity* [36, 26]. The mode connectivity hypothesis states that given two points in the landscape, there exists a path connecting them such that the loss is constant or near constant (or, said differently, the loss barrier is null). We refer to *linear mode connectivity* when the path connecting the two solutions is linear [32]. Recently, evidence has surfaced that stochastic gradient descent (SGD) solutions to the loss minimization problem can be linearly connected. Indeed, Entezari et al. [30] discuss the role of permutation symmetries from a loss connectivity viewpoint, conjecturing the possibility that mode connectivity is actually linear once accounting for all permutation invariances.

37th Conference on Neural Information Processing Systems (NeurIPS 2023).

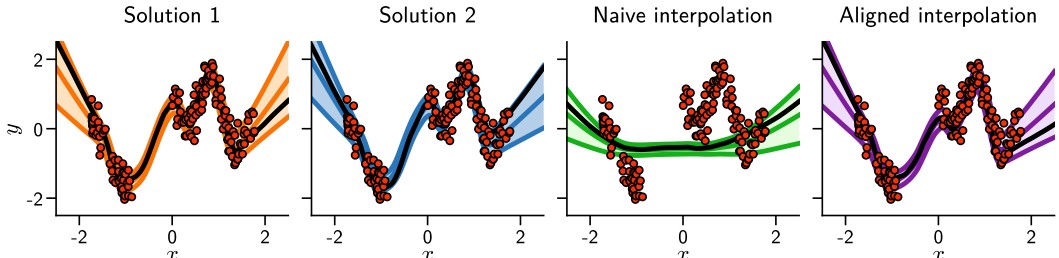

*Figure 1:* **Permutation symmetries for regression on the Snelson dataset [95].** *(Left)* Two different solutions, with similar function-space behavior (showing $\mu \pm 2\sigma$). *(Right)* Functions obtained when using two different strategies to interpolate between solutions. When we align the solutions, by taking into account the permutation symmetries, we retain the capability to model the data, indicating that there are solutions that once linearly interpolated exhibit no loss in performance. Note that, in weight space, the solution found with alignment is neither equal to solution 1 nor 2 (see the black curve which is a single function sample with fixed randomness).

Additionally, [2] gathers compelling empirical evidence across several network architectures and tasks, that under such a permutation symmetry the loss landscape often contains a single, nearly convex basin.

In this work, we are taking a different perspective on this analysis. We are interested in the Bayesian treatment of neural networks, which results in a natural form of regularization and allows to reason about uncertainty in the predictions [101, 73, 66]. Bayesian inference for deep neural networks is notoriously challenging, as we wish to marginalize over multi-modal distributions with high dimensionality [47]. For this reason, there are various ways to approximate the posterior, involving techniques like variational inference [39, 14, 35, 62, 77], Markov chain Monte Carlo (MCMC) methods [75, 72, 27], possibly with stochastic gradients [19, 112, 33, 105, 70] and the Laplace approximation [68, 68, 84]. Indeed, fundamentally the Bayesian posterior and the loss landscapes are tightly interconnected: (i) solutions to the loss minimization problem are equivalent to maximum-a-posteriori (MAP) solutions, (ii) the loss landscape is equivalent to the un-normalized negative log-posterior. While in theory, given a dataset, the posterior is unique and the solution is global, many approximations will only explore local properties of the true posterior[1]. It's worth noting that a posterior over the parameters of the neural network induces a posterior on the functions generated by the model. Permutation symmetries play an important role in the geometry of the weight-space posterior, which are generally not reflected in function-space. While it is possible to carry out inference directly in function space, this poses a number of challenges [99, 71, 89, 61]. Fig. 1 illustrates this situation for a regression task on the Snelson dataset using a 3-layer DNN: on the left we compare two (approximate) solutions which have different weight-space posterior but similar function-space behavior. Notably, when we interpolate these two solutions (Fig. 1 on the right), we completely lose all capability of modeling the data. However, when we account for permutation symmetries in the posterior, we end up with solutions that once interpolated are still good approximations. This suggests that for any weight-space distribution, there exists a class of solutions which are functionally equivalent and linearly connected. This example motivates an informal generic conjecture:

*Solutions of approximate Bayesian inference for neural networks are linearly connected after accounting for functionally equivalent permutations.*

While being similar to the one in [30, 2], if this conjecture was to hold true for approximate Bayesian neural networks (BNNs) it would represent an important step in further characterizing the properties of the Bayesian posterior and the effect of various approximations. We purposely leave the previous conjecture broadly open regarding the choice of the approximation method to allow for a more general discussion. More specifically, in this paper we will analyze and focus our discussion on the variational inference framework, making a more specific conjecture:

**Conjecture 1.** *Variational inference solutions for approximate Bayesian inference in neural networks are linearly connected after accounting for functionally equivalent permutations.*

---

[1]By local properties, we mean that despite theoretical convergence guarantees of many methods like variational inference and MCMC, in practice the true posterior for deep neural networks is still highly elusive; see for instance the empirical convergence analysis of Hamiltonian Monte Carlo (HMC) in [47].

**Contributions.** With this work, we aim at studying the linear connectivity properties of approximate solutions to the Bayesian inference problem and we make several contributions. (i) We extend the formalism of loss barrier and solution interpolation to BNNs. (ii) For the variational inference setting, propose a matching algorithm to search for linearly connected solutions by aligning the distributions of two independent solutions with respect to permutation matrices. Inspired by [2], we frame the problem as a combinatorial optimization problem using approximation to the linear sum assignment problem. (iii) We then experiment on a variety of architectures and datasets, finding nearly zero-loss barriers for linearly connected solutions. In Fig. 2 we present a sneak-peek and a visualization of our findings, where we show that after weight distribution alignment we can find a permutation map $P$ of the solution $q_1$ such that it can be linearly connected through high density regions to $q_0$.

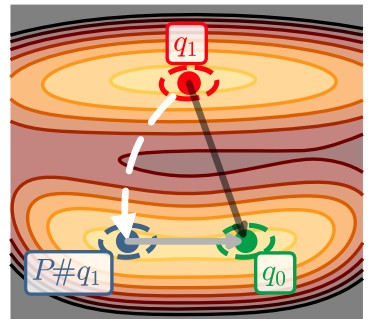

Log-posterior for CIFAR10

*Figure 2:* **Permutations in multi-modal posterior.** Log-posterior for MLP/CIFAR10, showing the two solutions ($q_0$ and $q_1$) for which we can find a permutation map such that $P_{\#}q_1$ can be linearly connected to $q_0$ with low barrier (brighter regions).

## 2 Preliminaries on Bayesian deep learning

In this section, we review some basic notations on BNNs and we review stochastic variational inference (SVI), which is the main approximation method that we analyze in this paper. Let's consider a generic multilayer perceptron (MLP) with $L$ layers, where the output of the $l$-th layer $\boldsymbol{f}_l(\boldsymbol{\theta}_l, \boldsymbol{x})$ is a vector-valued function of the previous layer output $\boldsymbol{f}_{l-1}$ as follows,

$$\boldsymbol{f}_l(\boldsymbol{\theta}_l, \boldsymbol{x}) = \boldsymbol{W}_l a(\boldsymbol{f}_{l-1}(\boldsymbol{\theta}_{l-1}, \boldsymbol{x})) + \boldsymbol{b}_l \tag{1}$$

where $a(\cdot)$ is a non-linearity, $\boldsymbol{W}_l$ is a $D_l \times D_{l-1}$ weight matrix and $\boldsymbol{b}_l$ the corresponding bias vector. We shall refer to the parameters of the layer $l$ as $\boldsymbol{\theta}_l = \{\boldsymbol{b}_l, \boldsymbol{W}_l\}$, and the union of all trainable parameters as $\boldsymbol{\theta} = \{\boldsymbol{\theta}_l\}_{l=1}^L$.

The objective of using Bayesian inference on deep neural networks [67, 65] involves inferring a posterior distribution over the parameters of the neural network given the available dataset $\{\boldsymbol{X}, \boldsymbol{Y}\} = \{(\boldsymbol{x}_i, \boldsymbol{y}_i)\}_{i=1}^N$. This requires choosing a likelihood and a prior [73, 74, 103]:

$$p(\boldsymbol{\theta} \,|\, \boldsymbol{Y}, \boldsymbol{X}) = Z^{-1} p(\boldsymbol{Y} \,|\, \boldsymbol{\theta}, \boldsymbol{X}) p(\boldsymbol{\theta}) \tag{2}$$

where the normalization constant $Z$ is the marginal likelihood $p(\boldsymbol{Y} \,|\, \boldsymbol{X})$. As usually done, we assume that the likelihood factorizes over observations, i.e. $p(\boldsymbol{Y} \,|\, \boldsymbol{\theta}, \boldsymbol{X}) = \prod_{i=1}^N p(\boldsymbol{y}_i \,|\, \boldsymbol{\theta}, \boldsymbol{x}_i)$.

Bayesian deep learning is intractable due to the non-conjugacy likelihood-prior and thus we don't have access to closed form solutions. Variational inference (VI) is a common technique to handle intractable Bayesian neural networks [12, 43, 39, 50]. Let $\mathcal{P}(\mathbb{R}^d)$ be the space of probability measures on $\mathbb{R}^d$; VI reframes the inference problem into an optimization one, commonly by introducing a parameterized distribution $q(\boldsymbol{\theta}) \in \mathcal{P}(\mathbb{R}^d)$ which is optimized to minimize the Kullback-Leibler (KL) divergence with respect to the true posterior $p(\boldsymbol{\theta} \,|\, \boldsymbol{Y}, \boldsymbol{X})$. In practice, this involves the maximization of the evidence lower bound (ELBO) defined as

$$\mathcal{L}_{\text{ELBO}}(q) \stackrel{\text{def}}{=} \int \log p(\boldsymbol{Y} \,|\, \boldsymbol{\theta}, \boldsymbol{X}) \mathrm{d}q(\boldsymbol{\theta}) - \text{KL}\left[q(\boldsymbol{\theta}) \,\|\, p(\boldsymbol{\theta})\right] \tag{3}$$

whose gradients can be unbiasedly estimated with mini-batches of data [43] and the reparameterization trick [54, 55]. Despite its simple formulation, the optimization of the ELBO hides several challenges, like the initialization of the variational parameters [87], the effects of over-parameterization on the quality of the approximation [88, 44, 58]. Here we are interested in how different solutions to Eq. (3) relate to each other in terms of loss barrier, which we will define formally in the following section.

## 3 Loss barriers

In the context of Bayesian inference, we are interested in the loss computed by marginalization of the model parameters with respect to (an approximation of) the posterior. As such, we use the predictive

likelihood, a proper scoring method for probabilistic models [83], defined as

$$\log p(\boldsymbol{y}^\star \mid \boldsymbol{x}^\star) = \log \int p(\boldsymbol{y}^\star \mid \boldsymbol{\theta}, \boldsymbol{x}^\star) p(\boldsymbol{\theta} \mid \boldsymbol{Y}, \boldsymbol{X}) \mathrm{d}\boldsymbol{\theta} \approx \log \int p(\boldsymbol{y}^\star \mid \boldsymbol{\theta}, \boldsymbol{x}^\star) \mathrm{d}q(\boldsymbol{\theta}) \tag{4}$$

where $q(\boldsymbol{\theta})$ is an approximation of the true posterior (parametric or otherwise), $\{\boldsymbol{x}^\star, \boldsymbol{y}^\star\}$ are respectively the input and its corresponding label a data point under evaluation. To keep the notation uncluttered for the remaining of the paper, we write the predictive likelihood computed for a set of points $\{\boldsymbol{x}_i^\star, \boldsymbol{y}_i^\star\}_{i=1}^N$ as a functional $\mathcal{L} : \mathcal{P}(\mathbb{R}^d) \to \mathbb{R}$, defined as

$$\mathcal{L}(q) \overset{\text{def}}{=} \sum_{i=1}^N \log \int p(\boldsymbol{y}_i^\star \mid \boldsymbol{\theta}, \boldsymbol{x}_i^\star) \mathrm{d}q(\boldsymbol{\theta}) \tag{5}$$

Let's assume two models trained with VI with two different initializations, random seeds, and batch ordering. Variational inference in the classic inverse sense $\mathrm{KL}\left[q \parallel p\right]$ is mode seeking, thus we expect the two runs to converge to different solutions, say $q_0$ and $q_1$. To test the loss barrier as we interpolate between the two solutions we need to decide on the interpolation rule. We decide to interpolate the solutions following the Wasserstein geodesics between $q_0$ and $q_1$. First, let's start with a few definitions. Let $q \in \mathcal{P}(\mathbb{R}^d)$ be a probability measure on $\mathbb{R}^d$ and $T : \mathbb{R}^d \to \mathbb{R}^d$ a measurable map; we denote $T_{\#}q$ the *push-forward measure* of $q$ through $T$. Now we can introduce the *Wasserstein geodesics*, as follows.

**Definition 1.** *The Wasserstein geodesics between $q_0$ and $q_1$ is defined as the path*

$$q_\tau = \left((1-\tau)\mathrm{Id} + \tau T_{q_0}^{q_1}\right)_{\#} q_0, \qquad \tau \in [0,1] \tag{6}$$

*where* $\mathrm{Id}$ *is the identity map and* $T_{q_0}^{q_1}$ *is the optimal transport map between $q_0$ and $q_1$, which for Brenier's theorem [16], is unique.*

While we could interpolate using a mixture of the two solutions, we argue that this choice is trivial and does not fully give us a picture of the underlying loss landscape. Indeed, Eq. (6) is fundamentally different from a naive mixture path $\tilde{q}_\tau = (1-\tau)q_0 + \tau q_1$. In case of Gaussian distributions, when $q_0 = \mathcal{N}(\boldsymbol{m}_0, \boldsymbol{S}_0)$ and $q_1 = \mathcal{N}(\boldsymbol{m}_1, \boldsymbol{S}_1)$, $q_\tau$ is Gaussian as well [100] with mean and covariance computed as follows:

$$\boldsymbol{m}_\tau = (1-\tau)\boldsymbol{m}_1 + \tau\boldsymbol{m}_2$$

$$\boldsymbol{S}_\tau = \boldsymbol{S}_1^{-1/2} \left((1-\tau)\boldsymbol{S}_1 + \tau\left(\boldsymbol{S}_1^{1/2}\boldsymbol{S}_2\boldsymbol{S}_1^{1/2}\right)^{1/2}\right)^2 \boldsymbol{S}_1^{-1/2} \tag{7}$$

which simplifies even further when the covariances are diagonal.

Now, we can define convexity along Wasserstein geodesics [4] as follows.

**Definition 2.** *Let $\mathcal{L} : \mathcal{P}(\mathbb{R}^d) \to \mathbb{R}$, $\mathcal{L}$ is $\lambda$ geodesics convex with $\lambda > 0$ if for any $q_0, q_1 \in \mathcal{P}(\mathbb{R}^d)$ it holds that*

$$\mathcal{L}(q_\tau) \leq (1-\tau)\mathcal{L}(q_0) + \tau\mathcal{L}(q_1) - \frac{\lambda\tau(1-\tau)}{2}\mathcal{W}_2^2(q_0, q_1) \tag{8}$$

*where $\mathcal{W}_2^2(q_0, q_1)$ is the Wasserstein distance defined as [104, 51, 52]*

$$\mathcal{W}_2^2(q_1, q_0) = \inf_{\gamma \in \Pi(q_1, q_0)} \int \|\boldsymbol{\theta}_1 - \boldsymbol{\theta}_0\|_2^2 \mathrm{d}\gamma(\boldsymbol{\theta}_1, \boldsymbol{\theta}_0) \tag{9}$$

*with $\Pi(\cdot, \cdot)$ being the space of measure with $q_0$ and $q_1$ as marginals.*

While mathematically proving the geodesics convexity of the predictive likelihood for arbitrary architectures and densities is currently beyond the scope of this work, we can empirically define a proxy using the *functional loss barrier*, defined as follows.

**Definition 3.** *The functional loss barrier along the Wasserstein geodesics from $q_0$ and $q_1$ is defined as the highest difference between the marginal loss computed when interpolating two solutions $q_0$ and $q_1$ and the linear interpolation of the loss at $q_0$ and $q_1$:*

$$\mathcal{B}(q_0, q_1) = \max_\tau \mathcal{L}(q_\tau) - \left((1-\tau)\mathcal{L}(q_0) + \tau\mathcal{L}(q_1)\right) \tag{10}$$

*where $q_\tau$ follows the definition in Eq. (6).*

This definition is a more general than the ones in [2, 30, 32] but we can recover [30] by assuming delta posteriors $q_i = \delta(\boldsymbol{\theta} - \boldsymbol{\theta}_i)$ and we can further recover [2, 32] by also assuming $\mathcal{L}(q_0) = \mathcal{L}(q_1)$.

**A comment on mixtures.** In previous paragraphs, we argued that the mixture of distributions is not sufficient to capture the underlying complex geometry of the posterior. Now, we want to better illustrate this choice with a simple example. In Fig. 3 we plot the test likelihood with two interpolation strategies between two solutions (MLP on CIFAR10): the Wasserstein geodesics and the mixture. With mixtures, we see that the likelihood is pretty much constant during the interpolation, but this is very miss-leading: we don't see barriers not because they don't exist, but because the mixture simply re-weights the distributions, without continuously transporting mass in the parameter space.

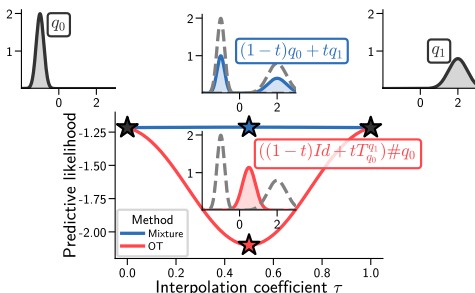

*Figure 3:* **Wasserstein geodesics and mixtures.** Test likelihood for mixture and the Wasserstein geodesics interpolation. Solutions are MLPs trained on CIFAR10.

# 4 Aligning distributions by looking for permutation symmetries

In this section, we formalize the algorithm that aligns the solutions of Bayesian inference through permutation symmetries of weight matrices and biases. Let $\mathbb{S}(d)$ be the set of valid $d \times d$ permutation matrices. Given a generic distribution $q(\boldsymbol{\theta})$, we can apply a permutation matrix $\boldsymbol{P} \in \mathbb{S}(D_l)$ to a hidden layer output at layer $l$, and if we define $\boldsymbol{\theta}'$ to be equivalent to $\boldsymbol{\theta}$ with the exception of

$$\boldsymbol{W}'_l = \boldsymbol{P}\boldsymbol{W}_l, \quad \boldsymbol{b}'_l = \boldsymbol{P}\boldsymbol{b}_l, \quad \boldsymbol{W}'_{l+1} = \boldsymbol{W}_{l+1}\boldsymbol{P}^\top, \tag{11}$$

then $P_{\#}q$ is the equivalent push-forward distribution for $\boldsymbol{\theta}'$, where $P$ is the associated permutation map. Let us define the distribution over the functional output of the model as

$$q(\boldsymbol{f}(\boldsymbol{\theta}, \cdot)) = \int \delta\left(\boldsymbol{f}(\boldsymbol{\theta}, \cdot) - \boldsymbol{f}(\widehat{\boldsymbol{\theta}}, \cdot)\right) \mathrm{d}q(\widehat{\boldsymbol{\theta}}), \tag{12}$$

and, equivalently, the distribution on the function using the permuted parameters as

$$q(\boldsymbol{f}(\boldsymbol{\theta}', \cdot)) = \int \delta\left(\boldsymbol{f}(\boldsymbol{\theta}', \cdot) - \boldsymbol{f}(\widehat{\boldsymbol{\theta}}', \cdot)\right) \mathrm{d}P_{\#}q(\widehat{\boldsymbol{\theta}}'), \tag{13}$$

where in both cases $\delta(\cdot)$ is the Dirac function. Then, it is simple to verify that the two models are functionally equivalent for any inputs,

$$q(\boldsymbol{f}(\boldsymbol{\theta}, \cdot)) = q(\boldsymbol{f}(\boldsymbol{\theta}', \cdot)). \tag{14}$$

This implies that for any weight-space distribution $q$, there exists a class of functionally equivalent solutions $P_{\#}q$, in the sense of Eq. (14). These same considerations can be easily extended to other layers, by considering multiple permutation matrices $\boldsymbol{P}_l$. For our analysis, given two solutions $q_0$ and $q_1$ we are interested in finding the permuted distribution $P_{\#}q_1$, functionally equivalent to $q_1$, in such a way that once interpolating using Eq. (6) we observe similar performance to $q_0$ and $q_1$. Formally, we can write

$$\arg\min_{P} \mathcal{D}(q_1(\boldsymbol{f}(\boldsymbol{\theta}', \cdot)), q_0(\boldsymbol{f}(\boldsymbol{\theta}, \cdot))) = \arg\min_{P} \mathcal{D}(P_{\#}q_1(\boldsymbol{\theta}), q_0(\boldsymbol{\theta})), \tag{15}$$

where $\mathcal{D}$ is a generic measure of discrepancy.[2]

## 4.1 Problem setup for permutation of vectors

We start from a single vector of parameters, disregarding for the moment the functional equivalence constraint. We will extend these results to matrices and multiple layers later. In practice considering

---

[2]To be formally correct, the l.h.s. is a discrepancy defined on stochastic processes while the r.h.s. is defined on random vectors. Under mild assumptions on the distribution on the parameters and the architectures, $\mathcal{D}$ is well defined in both cases [see e.g., 81].

Gaussian distributions, we know that if $q = \mathcal{N}(\boldsymbol{m}, \boldsymbol{S})$, then $P_{\#}q = \mathcal{N}(\boldsymbol{Pm}, \boldsymbol{PSP}^{\top})$ and if $q = \mathcal{N}(\boldsymbol{m}, \mathrm{diag}(\boldsymbol{s}^2))$ then $P_{\#}q = \mathcal{N}(\boldsymbol{Pm}, \mathrm{diag}(\boldsymbol{Ps}^2))$ [11]. With the KL divergence $\mathrm{KL}\left[P_{\#}q_1 \parallel q_0\right]$ it's easy to verify that it leads to just a distance between means, disregarding any covariance information. While certainly this represents a valid choice, we argue that we can find a better solution by using the Wasserstein distance. For Gaussian measures, the Wasserstein distance has analytic solution:

$$\mathcal{W}_2^2(q_1, q_0) = \|\boldsymbol{m}_0 - \boldsymbol{m}_1\|_2^2 + \mathrm{Tr}\left(\boldsymbol{S}_1 + \boldsymbol{S}_0 - 2\left(\boldsymbol{S}_1^{1/2}\boldsymbol{S}_0\boldsymbol{S}_1^{1/2}\right)^{1/2}\right) =$$

$$= \|\boldsymbol{m}_0 - \boldsymbol{m}_1\|_2^2 + \|\boldsymbol{S}_0 - \boldsymbol{S}_1\|_F^2, \tag{16}$$

where $\|\cdot\|_F$ denotes the Frobenius norm, $\|\boldsymbol{A}\| = \sum_{ij} a_{ij}^2$, and where the second line is valid only if the covariances commute ($\boldsymbol{S}_1 \boldsymbol{S}_0 = \boldsymbol{S}_0 \boldsymbol{S}_1$). In our case, then, we can simplify as follows:

$$\mathcal{W}_2^2(P_{\#}q_1, q_0) = \|\boldsymbol{m}_0 - \boldsymbol{Pm}_1\|_2^2 + \|\boldsymbol{s}_0 - \boldsymbol{Ps}_1\|_2^2. \tag{17}$$

To summarize, the problem now can be written as:

$$\underset{\boldsymbol{P} \in \mathbb{S}(d)}{\arg\min} \|\boldsymbol{m}_0 - \boldsymbol{Pm}_1\|_2^2 + \|\boldsymbol{s}_0 - \boldsymbol{Ps}_1\|_2^2 = \underset{\boldsymbol{P} \in \mathbb{S}(d)}{\arg\max} \left\langle \boldsymbol{P} \middle| \boldsymbol{m}_0\boldsymbol{m}_1^{\top} + \boldsymbol{s}_0\boldsymbol{s}_1^{\top} \right\rangle_F, \tag{18}$$

where the expression $\langle \boldsymbol{A}|\boldsymbol{B} \rangle_F$ is the Frobenius inner product, $\langle \boldsymbol{A}|\boldsymbol{B} \rangle_F = \sum_{ij} A_{ij} B_{ij}$. Note that the r.h.s. of Eq. (18) is a valid instantiation of the linear assignment problem (LAP) [10], which can be solved in polynomial time.

## 4.2 From vectors to neural network parameters

Finally, we need to take into account that we have multiple layers and weight matrices, and that we are trying to find functionally equivalent solutions. For this, we decide to explicitly change our main objective by enforcing the functional equivalence constraint as follows:

$$\underset{\{P_i\}}{\arg\min} \mathcal{W}_2^2\left(P_{1\#}q_1^{(1)}, q_0^{(1)}\right) + \mathcal{W}_2^2\left((P_2 \circ P_1^{\top})_{\#}q_1^{(2)}, q_0^{(2)}\right) + \cdots + \mathcal{W}_2^2\left((P_{L-1}^{\top})_{\#}q_1^{(L)}, q_0^{(L)}\right),$$

where the notation $\left(P_l \circ P_{l-1}^{\top}\right)$ represents the composition of the two permutation maps applied to rows and columns of the random weight matrices. More conveniently, this can be rewritten in terms of means and standard deviations. To leave the notation uncluttered, let's collect the means and the standard deviations for the layer $l$ in $\boldsymbol{M}^{(l)}$ and $\boldsymbol{S}^{(l)}$, which are now both $D_l \times D_{l-1}$ matrices, so that $q^{(l)} = \prod_{ij} \mathcal{N}(M_{ij}^{(l)}, S_{ij}^{(l)})$. Now we can write,

$$\underset{\{\boldsymbol{P}_i\}_{i=1}^L}{\arg\max} \left\langle \boldsymbol{M}_0^{(1)} \middle| \boldsymbol{P}_1 \boldsymbol{M}_1^{(1)} \right\rangle_F + \left\langle \boldsymbol{S}_0^{(1)} \middle| \boldsymbol{P}_1 \boldsymbol{S}_1^{(1)} \right\rangle_F + \left\langle \boldsymbol{M}_0^{(2)} \middle| \boldsymbol{P}_2 \boldsymbol{M}_1^{(2)} \boldsymbol{P}_1^{\top} \right\rangle_F + \left\langle \boldsymbol{S}_0^{(2)} \middle| \boldsymbol{P}_2 \boldsymbol{S}_1^{(2)} \boldsymbol{P}_1^{\top} \right\rangle_F +$$

$$+ \cdots + \left\langle \boldsymbol{M}_0^{(L)} \middle| \boldsymbol{M}_1^{(L)} \boldsymbol{P}_{L-1}^{\top} \right\rangle_F + \left\langle \boldsymbol{S}_0^{(L)} \middle| \boldsymbol{S}_1^{(L)} \boldsymbol{P}_{L-1}^{\top} \right\rangle_F.$$

This optimization problem is more challenging than the one presented in Eq. (18): we are interested in finding permutation matrices to be applied concurrently to rows and columns of both means and standard deviations. This class of problems, also known as sum of bilinear assignment problems (SOLAP), is NP-hard and no polynomial-time solutions exist. For this reason, we propose to use the setup in Ainsworth et al. [2] by extending it to our problem. In particular, by fixing all matrices with the exception of $\boldsymbol{P}_l$, we observe that also in our case the problem can be reduced to a classic LAP.

$$\underset{\boldsymbol{P}_l}{\arg\max} \left\langle \boldsymbol{M}_0^{(l)} \middle| \boldsymbol{P}_l \boldsymbol{M}_1^{(l)} \boldsymbol{P}_{l-1}^{\top} \right\rangle_F + \left\langle \boldsymbol{M}_0^{(l+1)} \middle| \boldsymbol{P}_{(l+1)} \boldsymbol{M}_1^{(l+1)} \boldsymbol{P}_l^{\top} \right\rangle_F +$$

$$\left\langle \boldsymbol{S}_0^{(l)} \middle| \boldsymbol{P}_l \boldsymbol{S}_1^{(l)} \boldsymbol{P}_{l-1}^{\top} \right\rangle_F + \left\langle \boldsymbol{S}_0^{(l+1)} \middle| \boldsymbol{P}_{(l+1)} \boldsymbol{S}_1^{(l+1)} \boldsymbol{P}_l^{\top} \right\rangle_F =$$

$$= \underset{\boldsymbol{P}_l}{\arg\max} \left\langle \boldsymbol{P}_l \middle| \boldsymbol{M}_0^{(l)} \boldsymbol{P}_{l-1} \left(\boldsymbol{M}_1^{(l)}\right)^{\top} + \left(\boldsymbol{M}_0^{(l+1)}\right)^{\top} \boldsymbol{P}_{l+1} \boldsymbol{M}_1^{(l+1)} + \right.$$

$$\left. \boldsymbol{S}_0^{(l)} \boldsymbol{P}_{l-1} \left(\boldsymbol{S}_1^{(l)}\right)^{\top} + \left(\boldsymbol{S}_0^{(l+1)}\right)^{\top} \boldsymbol{P}_{l+1} \boldsymbol{S}_1^{(l+1)} \right\rangle_F. \tag{19}$$

As discussed in [2], going through each layer, and greedily selecting its best $\boldsymbol{P}_l$, leads to a coordinate descent algorithm which guarantees to end in finite time. We present a pseudo-code in Algorithm 1.

**Algorithm 1:** Algorithm to align variational inference solutions

**Data:** Variational inference solutions $q_0$ and $q_1$
**Result:** Permutation matrices $\boldsymbol{P}_i$

1   $\boldsymbol{P}_i \leftarrow \boldsymbol{I}, \forall i \in \{1, \ldots, L\};$
2   **while** *not converged* **do**
3      **for** $i \in RandomPerm(1, \ldots, L-1)$ **do**
4         $\boldsymbol{P}_i = \arg\max \langle \boldsymbol{P}_i | \star \rangle$, where $\star$ is the r.h.s. of Eq. (19)

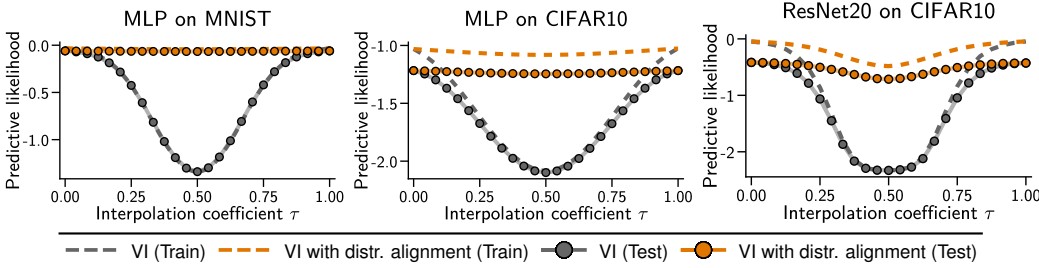

*Figure 4:* **Zero barrier solutions.** Comparison of loss barriers for standard VI (gray) and VI with alignment (orange). While loss barriers always appear between two solutions in the standard VI approach, in the case of VI with alignment there is no noticeable loss barrier for MLPs and a nearly-zero loss barrier for ResNet20.

## 5   Experiments

Now, we present some supporting evidence to Conjecture 1. We start by training two replicas of BNNs with variational inference (we refer to the Appendix for additional details on the experimental setup). We then compute the marginalized barrier as $\mathcal{B}(q_0, q_1) = \max_\tau \mathcal{L}(q_\tau) - ((1-\tau)\mathcal{L}(q_0) + \tau\mathcal{L}(q_1))$ where $\mathcal{L}(\cdot)$ is the predictive likelihood and $\tau \in [0, 1]$, from which we take 25 evenly distributed points. In particular, we seek to understand what happens to the VI solutions first for the naive interpolation from $q_0$ and $q_1$, and then for the interpolation after aligning $q_0$ and $P_\#q_1$. We experiment with MLPs with three layers and ResNet20 [41] with various widths on MNIST [60], Fashion-MNIST [108] and CIFAR10 [56]. All models are trained without data augmentation [106] and with filter response normalization (FRN) layers instead of BatchNorm. Finally, we set the prior to be Gaussian $\mathcal{N}(\boldsymbol{0}, \alpha^2\boldsymbol{I})$, with the flexibility of choosing the variance.

### 5.1   Low-barrier interpolations

Fig. 4 shows the results with and without alignment. We see that regardless of the dataset and the model used, the performance degrades significantly when we move between the two solutions with the naive interpolation, showing the existence of barriers in the predictive likelihood for Gaussian VI solutions. However, with the alignment proposed in § 4 and Algorithm 1, we recover zero barrier solutions for MLPs on both MNIST and CIFAR10, and nearly-zero barrier for ResNet20 on CIFAR10. This holds both for the train and test splits, with quantifiably smaller barriers in the test set.

In Fig. 5 we study the effect of the width of a neural network in relation to the loss barrier by taking an MLP and a ResNet20 with an increasing number of hidden features. We see that wider models generally provide lower barriers: for MLPs this holds true with and without alignment, while for the ResNet20 this is happening only after alignment. This extends some previous analysis done on loss-optimized networks. Specifically, Entezari et al. [30] show that barriers seem to have a double descent trend, while Ainsworth et al. [2] discuss that low barrier solutions after accounting for symmetries are easier to find in wider networks.

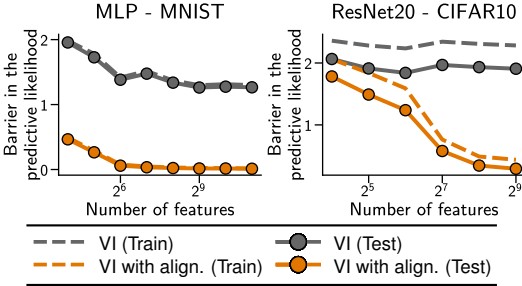

*Figure 5:* **Effect of width.** After distribution alignment, wider models exhibit lower likelihood barrier.

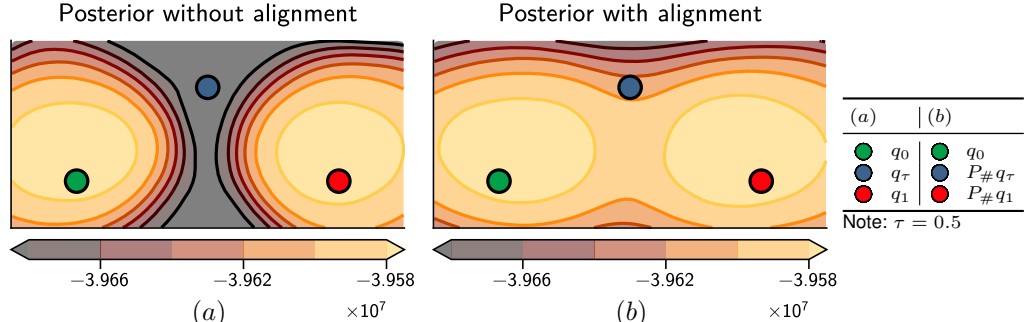

*Figure 6:* **Posterior density visualization.** Analysis of the log-posterior computed for ResNet20. Samples from $q_0$ and $q_1$ are connected by lower density regions, while $q_0$ and $P_\# q_1$ are not.

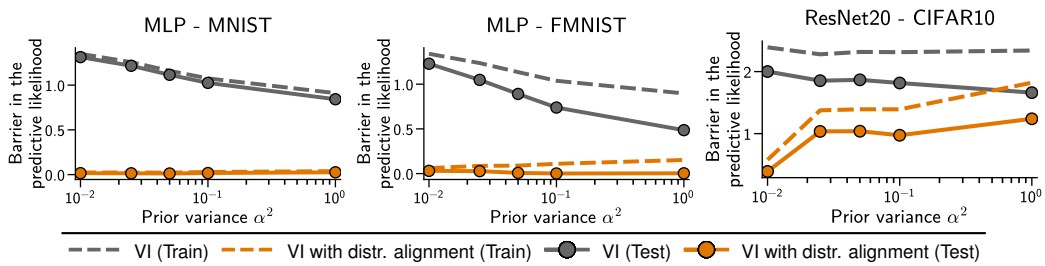

*Figure 7:* **Effect of prior variance.** After distribution alignment, prior variance has low effect in finding zero-loss barriers, while with naive interpolation we see a decreasing trend the higher the variance is.

We speculate that this might be due to the limiting behavior of Bayesian neural networks, which makes the posterior landscape Gaussian-like [48, 40]. While this does not fully explain the phenomenon observed, the existing connections between BNNs and non-parametric models, like Gaussian Processes (GPs) [83] and deep Gaussian processes (DGPs) [22, 20, 28, 93], can provide additional insights on the role of symmetries in weight space [80].

Finally, as an additional check, we analyze the log-posterior with and without alignment by projecting the density into two dimensional slices, following the setup in [47, 36]. We study the two dimensional subspace of the parameter space supported by the hyperplane $H = \{\boldsymbol{\theta} \in \mathbb{R}^d \,|\, \boldsymbol{\theta} = a\boldsymbol{\theta}_a + b\boldsymbol{\theta}_b + (1 - a - b)\boldsymbol{\theta}_c\}$, where $a, b \in \mathbb{R}$ and $\boldsymbol{\theta}_a$, $\boldsymbol{\theta}_b$ and $\boldsymbol{\theta}_c$ are the samples either from $q_0$, $q_1$ and $q_\tau$ without alignment or from $q_0$, $P_\# q_1$ and $P_\# q_\tau$ with alignment. With this configuration, all three samples always lie on this hyper-plane. In Fig. 6, we present the visualization of ResNet20 trained on CIFAR10. We see that the samples from $q_0$ and $P_\# q_1$ are connected by higher density regions than the ones between $q_0$ and $q_1$. This is in line with the results in Fig. 4, where we see that the loss barrier is lower after alignment.

## 5.2 Analyzing the effect of the prior and testing the cold posterior effect

In all previous experiments we used a Gaussian prior $\mathcal{N}(\mathbf{0}, \alpha^2 \boldsymbol{I})$ with fixed $\alpha^2$; now we study the effect of a varying prior variance $\alpha^2$ on the behavior of the loss barriers. We experiment this on a MLP trained on MNIST and Fashion-MNIST and on a ResNet20 (width x8) on CIFAR10. We report the results in Fig. 7. We can appreciate two behaviors: with alignment, there is no measurable effect of using different variances in finding zero-barrier solutions; on the contrary, without alignment we see that naive VI solutions are easier to interpolate with lower barrier when the prior is more diffused. At the same time, we see that higher variances produce bigger gaps between train barriers and test barriers. We speculate that this is due to overfitting happening with more relaxed priors, which makes low-barrier (but low-likelihood) solutions easier to find.

Additionally, several previous works have analyzed the effect of tempering the posterior in BNNs [106, 112, 110, 47]. Specifically, we are interested in the distribution $p_T(\boldsymbol{\theta} \,|\, \boldsymbol{Y}) \propto (p(\boldsymbol{Y} \,|\, \boldsymbol{\theta}, \boldsymbol{X}) p(\boldsymbol{\theta}))^{1/T}$, where $T$ is known as the temperature. Note that starting from the above definition, we can write an equivalent ELBO for VI which takes into account $T$. For $T < 1$, we have cold posteriors, which are

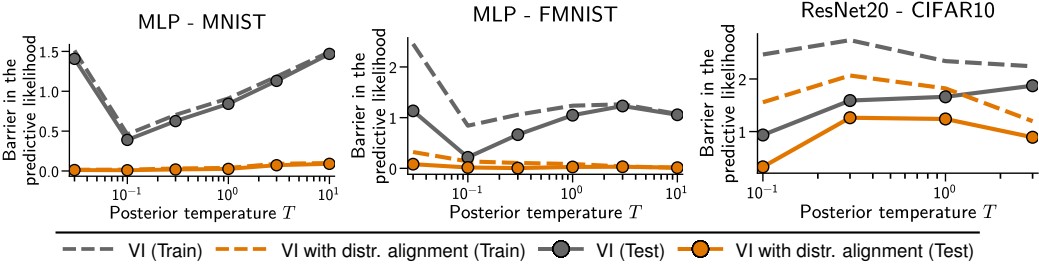

*Figure 8:* **Effect of temperature.** After alignment, cold posteriors makes barriers marginally closer to zero

sharper than the true Bayesian posteriors, while for $T > 1$ we have warm posterior, which are more diffused. In Fig. 8 we see that barriers for cold posteriors with alignment are marginally closer to zero than for warm posteriors. Note that cold temperatures concentrate the distribution around the MAP, which motivates a further comparison with a non-Bayesian approach.

## 5.3 Comparison with SGD solutions

Motivated by the results with cold posteriors, we also compare the behavior of barriers for VI versus MAP solutions obtained via SGD. Note that using the MAP solution we end up with the same setup of weight matching than in [2]. In Fig. 9 we report this comparison, which is carried out on ResNet20 for various width multipliers. For narrow models, we see that VI with alignment and MAP with weight matching both behave in a similar manner. Interestingly, as the model becomes wider, MAP solutions achieve marginally lower barriers than VI. We speculate that this might be due to the simple Gaussian parameterization for the approximate posterior, which doesn't completely capture the local geometry of the true posterior.

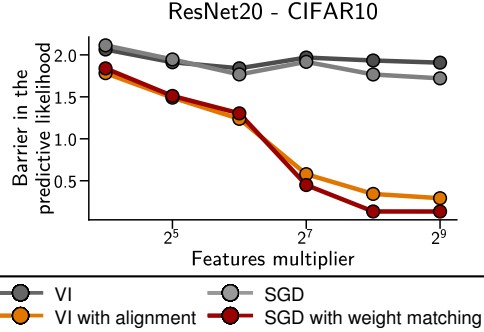

*Figure 9:* **VI versus SGD.** VI and SGD behave equivalently after permutation alignment in narrow models. For wider models, SGD solutions reach lower barriers than VI.

## 5.4 Effect of normalization layers and data augmentation

We conclude this section with a discussion on the effects of normalization layers and data augmentation on the loss barriers for Bayesian neural networks.

Different normalization strategies can affect the overall geometry of the problem [30, 2]. As discussed in [49], interpolating with BatchNorm layers [45] is pathological due to *variance collapse* of the feature representation in hidden layers. Additionally, batch-dependent normalization layers don't have a clear Bayesian interpretation, since the likelihood cannot factorize. For our experiments we choose to use the FRN layer, as done in previous works [e.g., 47]. Note that FRN layers are invariant to permutation units and therefore can be aligned without problems. To test the *variance collapse* behavior, we analyze the variance of activations following the instructions in [49, §3.1], with the sole difference that the activations are marginalized w.r.t. samples from the posterior. In Fig. 10 we can see that there isn't a pathological variance collapse after alignment. Finally, in Fig. 11 we compare another normalization layer, the LayerNorm (LN) [6]. Note that LN is also batch-independent, it has a clear Bayesian interpretation and it is invariant to permutation units. Indeed, we see that both normalization layers can be aligned, and LN exhibits lower barriers than FRN.

Finally, in all previous experiments we skipped data augmentation, because the random augmentations introduce stochasticity which lacks a proper Bayesian interpretation in the formulation of the likelihood function (e.g. re-weighting of the likelihood due to the increase of the effective sample size [77]). Additionally, data augmentation can contribute to spurious effects difficult to disentangle (e.g., cold posterior effect [106]). Nonetheless, during the development of the method we didn't make an

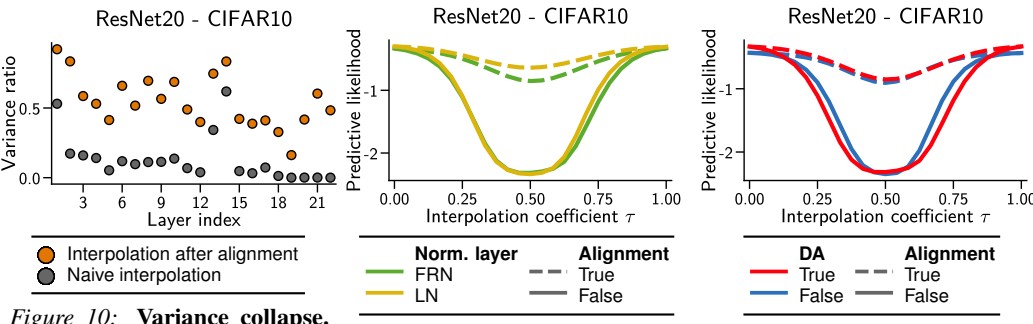

*Figure 10:* **Variance collapse.** Variance collapse is present for the naive interpolation, but it is reduced after alignment.

*Figure 11:* **Effect of normalization layers.** Both FRN and LN can be aligned, and LN exhibits lower barriers.

*Figure 12:* **Effect of data augmentation.** With and without DA, we are able to recover similar low barrier solutions.

assumption on data augmentation and in Fig. 12 we experiment both with and without augmentation, showing that we are still able to recover similar low barrier solutions in both cases. Having said that, we advocate caution when using data augmentation in Bayesian neural networks, as it changes the shape of the posterior.

# 6    Related work

In earlier sections of the paper, we already briefly discussed and reviewed relevant works on mode connectivity, symmetries in the loss landscape and connection with gradient-based optimization methods. Here we discuss some relevant works on the connection to Bayesian deep learning. The work of Garipov et al. [36] sparked several contributions on exploiting mode connectivity for ensembling models, which is akin to Bayesian model averaging. For example, in [46] the authors propose to ensemble models using curve subspaces to construct low-dimensional subspaces of parameter space. These curve subspaces are, among others, the non-linear paths connecting low-loss modes (and consequently high-posterior density) in weight space. In [31], the authors attempt to explain the effectiveness of deep ensembles [59], concluding that it is partially due to the diversity of the SGD solutions in parameter space induced by random initialization. More recently, in [9] the authors reason about mode connecting volumes, which are multi-dimensional manifolds of low loss that connect many independently trained models. These mode connecting volumes form the basis for an efficient method for building simplicial complexes for ensembling. Here, we want to highlight that these works have not taken into account the permutation symmetries. Finally, in a concurrent submission, [107] proposes an algorithm to remove symmetries in MCMC chains for tanh networks.

# 7    Conclusions

By studying the effect of permutation symmetries, which are ubiquitous in neural networks, it is possible to analyze the fundamental geometric properties of loss landscapes like (linear) mode connectivity and loss barriers. While previously this was done on loss-optimized networks [2, 30, 32], in this work we have extended the analysis to Bayesian neural networks. We have studied the linear connectivity properties of approximate Bayesian solutions and we have proposed a matching algorithm (Algorithm 1) to search for linearly connected solutions, by aligning the distributions of two independent VI solutions with respect to permutation matrices. We have empirically validated our framework on a variety of experiments, showing that we can find zero barrier linearly-connected solutions for BNNs trained with VI, on shallow models as well as on deep convolutional networks. This brings evidence for Conjecture 1 regarding the linear connectivity of variational inference approximations for BNNs. Furthermore, we have studied the effect of various design hyper-parameters, like width, prior and temperature, and observed complex patterns of behavior, which would require additional research. In particular, the experiments raise questions regarding the relation between linear mode connectivity and the generalization of BNNs, as well as the role of width with respect to limiting behaviors of non-parametric models like GPs and DGPs.

## Acknowledgments and Disclosure of Funding

The authors want to thank David Bertrand and the AIAO team at Stellantis for their work on the software and hardware infrastructure used for this work.

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

## A  Future work and open questions

Besides the specific open questions discussed above, we anticipate some possible future work. The framework we proposed to analyze the loss barriers in BNNs is general and can be applied to approximations other than VI. Future work could undertake to extend this analysis to the Laplace approximation [68]. However, this raises methodological challenges since Algorithm 1 is not directly applicable, due to the use of a dense covariance matrix (the inverse of the Hessian). Additionally, it would be worth extending these results to sample-based inference, like stochastic gradient Hamiltonian Monte Carlo (SGHMC) [19], or particle-based inference, like Stein variational inference [63]. This would call for a careful analysis, starting from the solution of Eq. (6), which would require approximations [21].

## B  Additional details on the alignment method

In the main paper, to align the distributions with respect to permutation matrices we argue to use the Wasserstein distance rather than the KL divergence. Indeed, by considering the KL divergence KL $[P_\# q_1 \parallel q_0]$ between Gaussians we have

$$\mathrm{KL}\left[P_\# q_1 \parallel q_0\right] = \log \det \mathrm{diag}(\boldsymbol{s}_0) - \log \det \mathrm{diag}(\boldsymbol{P}\boldsymbol{s}_1) + \mathrm{Tr}\big(\mathrm{diag}(\boldsymbol{P}\boldsymbol{s}_1\boldsymbol{s}_0^{-1})\big) + \quad (20)$$

$$(\boldsymbol{m}_0 - \boldsymbol{P}\boldsymbol{m}_1)^\top \mathrm{diag}(\boldsymbol{s}_0^{-1})(\boldsymbol{m}_0 - \boldsymbol{P}\boldsymbol{m}_1) \quad (21)$$

It's easy to verify that the first three terms do not depend on $\boldsymbol{P}$, leading to just a distance between means and disregarding any covariance information. In the figure below, we visualize the difference between doing LAP with the KL cost and LAP with the Wasserstein cost.

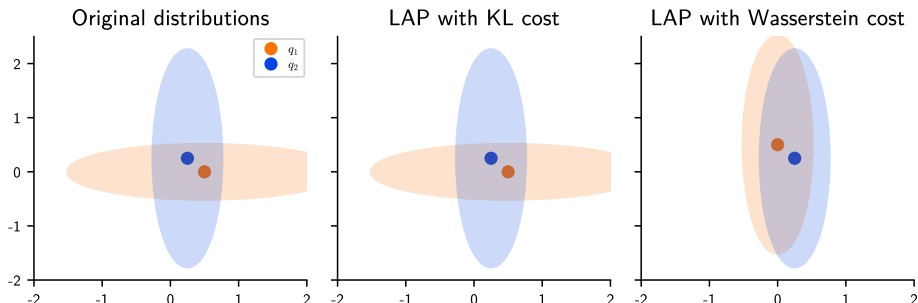

*Figure A13:* **Alignment using different objectives.** Given two distributions symmetrical w.r.t. the $y = x$ plane, using the KL cost LAP results in the identity permutation (which fails to recover the symmetry), while the Wasserstein cost better aligns the two distributions

## C  Experimental setup

If not otherwise stated, we start by training two replicas of BNNs with variational inference and we compute the marginalized barrier as $\mathcal{B}(q_0, q_1) = \max_\tau \mathcal{L}(q_\tau) - ((1-\tau)\mathcal{L}(q_0) + \tau\mathcal{L}(q_1))$ where $\mathcal{L}(\cdot)$ is the predictive likelihood and $\tau \in [0, 1]$, from which we take 25 evenly distributed points. In particular, we seek to understand what happens to the VI solutions with and without alignment applied to one of the two distribution ($q_1$ in our case). We experiment with MLPs with three layers and ResNet20 [41] with various widths on MNIST [60], Fashion-MNIST [108] and CIFAR10 [56]. All models are trained without data augmentation, because the random augmentations introduce stochasticity which lacks a proper Bayesian interpretation in the formulation of the likelihood function [106]. Finally, we set the prior to be Gaussian $\mathcal{N}(\boldsymbol{0}, \alpha^2\boldsymbol{I})$, with the flexibility of choosing the variance. All VI models are trained using the classic SGD optimizer with momentum [85, 91] using the reparameterization trick [54] with one sample during training and 128 samples during testing We use the categorical distribution and the Gaussian distribution as classification and regression likelihood, respectively. Tables 2 and 3 show details on the MLPs and convolutional neural networks (CNNs) base architectures used in our experimental campaign, while Table 1 reports the hyperparameters used in the experiments. Note that differently from Entezari et al. [30] and Ainsworth

*Table 1:* Hyperparameters used for the experiments

| Dataset | CIFAR10 | | MNIST |
|---|---|---|---|
| **Model** | ResNet20 | MLP | MLP |
| Data Aug. | False | False | False |
| Batch size | 500 | 500 | 500 |
| Temperature | 1.0 | 1.0 | 1.0 |
| Test samples | 128 | 128 | 128 |
| Train samples | 1 | 1 | 1 |
| VI std. init | 0.01 | 0.01 | 0.01 |
| Base features | 16 | 512 | 512 |
| Prior var | 0.01 | 0.0025 | 0.01 |
| Learning rate | 0.000001 | 0.000001 | 0.000001 |
| Train epochs | 1000 | 1000 | 1000 |

et al. [2], we don't use data augmentation. A possible protocol for handling data augmentation in BNNs is presented by Osawa et al. [77] and involves carefully tuning the likelihood temperature to correctly counting the number of data points.

*Table 2:* MLP

| Layer | Dimensions |
|---|---|
| Linear-ReLU | $512 \times D_{in}$ |
| Linear-ReLU | $512 \times 512$ |
| Linear-ReLU | $512 \times 512$ |
| Linear-Softmax | $D_{out} \times 512$ |

*Table 3:* ResNet20

| Layer | Dimensions |
|---|---|
| Conv2D | $16 \times 3 \times 3 \times D_{in}$ |
| Residual Block | $\begin{bmatrix} 3 \times 3, 16 \\ 3 \times 3, 16 \end{bmatrix} \times 3$ |
| Residual Block | $\begin{bmatrix} 3 \times 3, 32 \\ 3 \times 3, 32 \end{bmatrix} \times 3$ |
| Residual Block | $\begin{bmatrix} 3 \times 3, 64 \\ 3 \times 3, 64 \end{bmatrix} \times 3$ |
| AvgPool | $8 \times 8$ |
| Linear-Softmax | $D_{out} \times 64$ |

### C.1 Computing platform

The experiments have been performed using JAX [15] and run on two AWS p4d.24xlarge instances with 8 NVIDIA A100 GPUs. Experiments were conducted using in the eu-west-1 region, which has a carbon efficiency of 0.62 kgCO$_2$eq/kWh. A cumulative of 6500 hours of computation was performed on GPUs and it includes interactive sessions as well as small experiments with very low GPU usage, providing a pessimistic estimation of the true utilization. Total emissions are estimated to be 1007.5 kgCO$_2$eq of which 100 percents were directly offset by AWS.

## D  Additional results

We present timings obtained by profiling the time needed to solve the SOLAP with the Wasserstein cost, as well as the time for the deterministic case [2]. In Fig. A14 we show the results for MLP and ResNet20 architectures, varying the model width. It is evident that, in the majority of cases, the algorithm completes within a minute. Moreover, as anticipated, in case of VI solving our distribution alignment problem for wide neural networks is more computationally demanding compared to merely matching weights from SGD solutions.

Finally, we also test our setup on the CIFAR100 dataset [56]. Surprisingly, we were not able to replicate the same level of performance as in the other cases. In Fig. A15, we see that, despite converging well, we fall short to find zero-barrier solutions. Similarly to the comments of Ainsworth et al. [2], we also stress that the failure to align distributions does not rule out the existence of a proper permutation map that the algorithm couldn't find. Nonetheless, this raises a number of questions: the Bayesian posterior is the product of two ingredients, the prior and the likelihood, conditioned to observing a dataset.

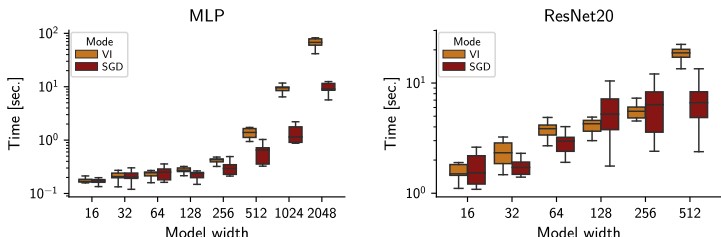

*Figure A14:* **Timings.** Profile of the algorithms to align the VI solutions and to match weights from SGD solutions.

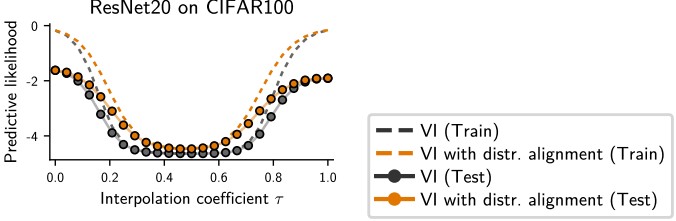

*Figure A15:* **Alignment failure.** The method proposed fails to recover zero-barrier solutions for CIFAR100.

Finally, as an additional check, we analyze the log-posterior by projecting the density into two dimensional slices, following the setup in [47, 36]. We study the two dimensional subspace of the parameter space supported by the hyperplane $H$ of the form

$$H = \left\{ \boldsymbol{\theta} \in \mathbb{R}^d \,|\, \boldsymbol{\theta} = a\boldsymbol{\theta}_a + b\boldsymbol{\theta}_b + (1 - a - b)\boldsymbol{\theta}_c \right\},$$

where $a, b \in \mathbb{R}$ and $\boldsymbol{\theta}_a$, $\boldsymbol{\theta}_b$ and $\boldsymbol{\theta}_c$ are the means of $q_0$, $q_1$ and $P_{\#}q_1$. With this configuration, all three solutions lie on this hyper-plane. In Fig. 6, we present the visualization of ResNet20 trained on CIFAR10. We see that the distributions $q_0$ and $P_{\#}q_1$ are connected by higher density regions than the ones between $q_0$ and $q_1$. Also, as expected the symmetries arise from the form of the likelihood, and the prior has a comparable strength with respect to the three posteriors. Later, we also study in more details the effect of the prior's variance in finding low-barrier solutions.

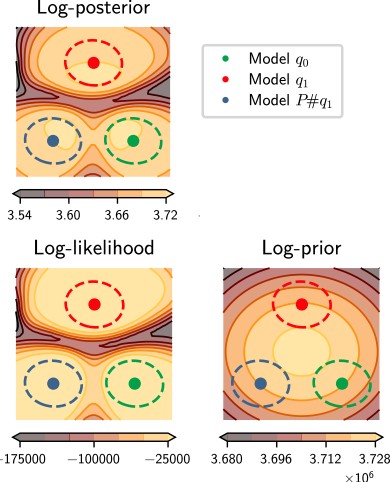

*Figure A16:* **Posterior density visualization.** All three solutions are local approximation of posterior, but $q_0$ and $P_{\#}q_1$ are connected by lower density regions.

## E A primer on variational inference for Bayesian neural networks

VI is a classic tool to tackle intractable Bayesian inference [50, 13]. VI casts the inference problem into an optimization-based procedure to compute a tractable approximation of the true posterior. Assume a generic parametric model $f$ parameterized by some unknown parameters $\boldsymbol{\theta}$ (i.e. $f(\cdot, \boldsymbol{\theta})$) and a collection of data $\boldsymbol{y} \in \mathbb{R}^N$ corresponding to some input points $\boldsymbol{X} = \{\boldsymbol{x}_i \,|\, \boldsymbol{x}_i \in \mathbb{R}^{D_{\text{in}}}\}_{i=1,\dots,N}$. In our setting, we have a probabilistic model $p(\boldsymbol{y} \,|\, f(\boldsymbol{X}; \boldsymbol{\theta}))$ with parameters $\boldsymbol{\theta}$, a prior distributions on them $p(\boldsymbol{\theta})$ and a set of observations $\{\boldsymbol{X}, \boldsymbol{y}\}$. In a nutshell, the general recipe of VI consists of (i) introducing a set $\mathcal{Q}$ of distributions; (ii) defining a tractable objective that "measure" the distance between any arbitrary distribution $q(\boldsymbol{\theta}) \in \mathcal{Q}$ and the true posterior $p(\boldsymbol{\theta} \,|\, \boldsymbol{y})$; and finally (iii) providing a programmatic way to find the distribution $\widetilde{q}(\boldsymbol{\theta})$ that minimizes such distance. In practice, $q(\boldsymbol{\theta})$ has some free parameters $\boldsymbol{\nu}$ (also known as *variational parameters*), which are optimized such that

the approximating distribution $q(\boldsymbol{\theta}; \boldsymbol{\nu})$ is as closer as possible to the true posterior $p(\boldsymbol{\theta} \,|\, \boldsymbol{y})$. We can derive the variational objective starting from the definition of the KL,

$$
\begin{aligned}
\mathrm{KL}\left[q(\boldsymbol{\theta}; \boldsymbol{\nu}) \,\|\, p(\boldsymbol{\theta} \,|\, \boldsymbol{y})\right] &= \mathbb{E}_{q(\boldsymbol{\theta}; \boldsymbol{\nu})}\left[\log q(\boldsymbol{\theta}; \boldsymbol{\nu}) - \log p(\boldsymbol{\theta} \,|\, \boldsymbol{y})\right] = \\
&= \mathbb{E}_{q(\boldsymbol{\theta}; \boldsymbol{\nu})}\left[\log q(\boldsymbol{\theta}; \boldsymbol{\nu}) - \log p(\boldsymbol{y} \,|\, \boldsymbol{\theta}) - \log p(\boldsymbol{\theta})\right] + \log p(\boldsymbol{y})
\end{aligned}
\tag{22}
$$

Rearranging we have that

$$
\log p(\boldsymbol{y}) - \mathrm{KL}\left[q(\boldsymbol{\theta}; \boldsymbol{\nu}) \,\|\, p(\boldsymbol{\theta} \,|\, \boldsymbol{y})\right] = \mathbb{E}_{q(\boldsymbol{\theta}; \boldsymbol{\nu})}\left[\log q(\boldsymbol{\theta}; \boldsymbol{\nu}) - \log p(\boldsymbol{y} \,|\, \boldsymbol{\theta}) - \log p(\boldsymbol{\theta})\right]
\tag{23}
$$

The r.h.s. of the equation defines our variational objective, also known as ELBO, that can be arranged as follows,

$$
\mathcal{L}_{\mathrm{ELBO}}(\boldsymbol{\nu}) = \underbrace{\mathbb{E}_{q(\boldsymbol{\theta}; \boldsymbol{\nu})} \log p(\boldsymbol{y} \,|\, \boldsymbol{\theta})}_{\text{Model fitting term}} - \underbrace{\mathrm{KL}\left[q(\boldsymbol{\theta}; \boldsymbol{\nu}) \,\|\, p(\boldsymbol{\theta})\right]}_{\text{Regularization term}}.
\tag{24}
$$

This formulation highlights the property of this objective, which is made of two components: the first one is the expected log-likelihood under the approximate posterior $q$ and measures how the model fits the data. The second term, on the other hand, has the regularization effect of penalizing posteriors that are far from the prior as measured by the KL. Before diving into the challenges of optimization of the ELBO, we shall spend a brief moment discussing the form of the approximating distribution $q$. One of the simplest and easier choice is the mean field approximation [42], where each variable $\theta_i$ is taken to be independent with respect to the remaining $\boldsymbol{\theta}_{-i}$. Effectively, this imposes a factorization of the posterior,

$$
q(\boldsymbol{\theta}; \boldsymbol{\nu}) = \prod_{i=1}^{K} q(\theta_i; \boldsymbol{\nu}_i)
\tag{25}
$$

where $\boldsymbol{\nu}_i$ is the set of variational parameters for the parameter $\theta_i$. On top of this approximation, $q(\theta_i)$ is often chosen to be Gaussian,

$$
q(\theta_i) = \mathcal{N}(\mu_i, \sigma_i^2)
\tag{26}
$$

Now, the collection of all means and variances $\{\mu_i, \sigma_i^2\}_{i=1}^{K}$ defines the set of variational parameters to optimize.

For BNNs the analytic evaluation of the ELBO (and its gradients) is always untractable due the non-linear nature of the expectation of the log-likelihood under the variational distribution. Nonetheless, this can be easily estimated via Monte Carlo integration [72], by sampling $N_{\mathrm{MC}}$ times from $q_{\boldsymbol{\nu}}$,

$$
\mathbb{E}_{q(\boldsymbol{\theta}; \boldsymbol{\nu})} \log p(\boldsymbol{y} \,|\, \boldsymbol{\theta}) \approx \frac{1}{N_{\mathrm{MC}}} \sum_{j=1}^{N_{\mathrm{MC}}} \log p(\boldsymbol{y} \,|\, \widetilde{\boldsymbol{\theta}}_j), \quad \text{with} \quad \widetilde{\boldsymbol{\theta}}_j \sim q(\boldsymbol{\theta}; \boldsymbol{\nu})
\tag{27}
$$

In practice, this is as simple as re-sampling the weights and the biases for all the layers $N_{\mathrm{MC}}$ times and computing the output for each new sample.

We now have a tractable objective that needs to be optimized with respect to the variational parameters $\boldsymbol{\nu}$. Very often the KL term is known, making its differentation trivial. On the other hand the expectation of the likelihood is not available, making the computation of its gradients more challenging. This problem can be solved using the so-called *reparameterization trick* [92, 54]. The reparameterization trick aims at constructing $\boldsymbol{\theta}$ as an invertible function $\mathcal{T}$ of the variational parameters $\boldsymbol{\nu}$ and of another random variable $\boldsymbol{\varepsilon}$, so that $\boldsymbol{\theta} = \mathcal{T}(\boldsymbol{\varepsilon}; \boldsymbol{\nu})$. Generally, a $\mathcal{T}$ that suits this constraint might not exists; Ruiz et al. [90] discuss how to build "weakly" dependent transformation $\mathcal{T}$ for distributions like Gamma, Beta and Log-normal. For discrete distributions, instead, one could use a continuous relaxation, like the Concrete [69]. $\boldsymbol{\varepsilon}$ is chosen such that its marginal $p(\boldsymbol{\varepsilon})$ does not depend on the variational parameters. With this parameterization, $\mathcal{T}$ separates the deterministic components of $q$ from the stochastic ones, making the computation of its gradient straightforward. For a Gaussian distribution with mean $\mu$ and variance $\sigma^2$, $\mathcal{T}$ corresponds to as simple scale-location transformation of an isotropic Gaussian noise,

$$
\theta \sim \mathcal{N}(\mu, \sigma^2) \iff \theta = \mu + \sigma \varepsilon \quad \text{with} \quad \varepsilon \sim \mathcal{N}(0, 1).
\tag{28}
$$

This simple transformation ensures that $p(\varepsilon) = \mathcal{N}(0, 1)$ does not depends on the variational parameters $\boldsymbol{\nu} = \{\mu, \sigma^2\}$. The gradients of the ELBO can be therefore computed as

$$
\boldsymbol{\nabla}_{\boldsymbol{\nu}} \mathcal{L}_{\mathrm{ELBO}} = \mathbb{E}_{p(\boldsymbol{\varepsilon})}\left[\boldsymbol{\nabla}_{\boldsymbol{\theta}} \log p(\boldsymbol{y} \,|\, \boldsymbol{\theta}) \,|\, _{\boldsymbol{\theta} = \mathcal{T}(\boldsymbol{\varepsilon}; \boldsymbol{\nu})} \boldsymbol{\nabla}_{\boldsymbol{\nu}} \mathcal{T}(\boldsymbol{\varepsilon}; \boldsymbol{\nu})\right] - \boldsymbol{\nabla}_{\boldsymbol{\nu}} \mathrm{KL}\left[q(\boldsymbol{\theta}; \boldsymbol{\nu}) \,\|\, p(\boldsymbol{\theta})\right].
\tag{29}
$$

The gradient $\boldsymbol{\nabla_\theta} \log p(\boldsymbol{y} \,|\, \boldsymbol{\theta})$ depends on the model and it can be derived with automatic differentation tools [1, 78], while $\boldsymbol{\nabla_\nu} \mathcal{T}(\varepsilon; \boldsymbol{\nu})$ doesn't have any stochastic components and therefore can be known deterministically. Note that the reparameterization trick can be also used when the KL is not analitically available. In that case, we would end up with,

$$\boldsymbol{\nabla_\nu} \mathcal{L}_{\mathrm{ELBO}} = \mathbb{E}_{p(\boldsymbol{\varepsilon})} \left[ \boldsymbol{\nabla_\theta} \log p(\boldsymbol{y} \,|\, \boldsymbol{\theta}) + \log q(\boldsymbol{\theta}; \boldsymbol{\nu}) - \log p(\boldsymbol{\theta}) \right]_{\boldsymbol{\theta} = \mathcal{T}(\boldsymbol{\varepsilon}; \boldsymbol{\nu})} \boldsymbol{\nabla_\nu} \mathcal{T}(\varepsilon; \boldsymbol{\nu}) \tag{30}$$

Roeder et al. [86] argue that when we believe that $q(\boldsymbol{\theta}; \boldsymbol{\nu}) \approx p(\boldsymbol{y} \,|\, \boldsymbol{\theta})$, Eq. (30) should be prefered over Eq. (29) even if computing analitically the KL is possible. Note that this case is very unlikely for BNN posteriors, and that the additional randomness introduced by the Monte Carlo estimation of the KL could be harmful.

In case of large datasets and complex models, the formulation summarized in Eq. (29) can be computationally challenging, due to the evaluation of the likelihood and its gradients $N_{\mathrm{MC}}$ times. Assuming factorization of the likelihood,

$$p(\boldsymbol{y} \,|\, \boldsymbol{\theta}) = p(\boldsymbol{y} \,|\, f(\boldsymbol{X}; \boldsymbol{\theta})) = \prod_{i=1}^{N} p(y_i \,|\, f(\boldsymbol{x}_i; \boldsymbol{\theta})) \tag{31}$$

this quantity can be approximated using mini-batching [39, 43]. Recalling $\boldsymbol{y}$ as the set of labels of our dataset with $N$ examples, by taking $\mathcal{B} \subset \boldsymbol{y}$ as a random subset of $\boldsymbol{y}$, the likelihood term can be estimated in an unbiased way as

$$\log p_{\boldsymbol{\theta}}(\boldsymbol{y} \,|\, \boldsymbol{\theta}) \approx \frac{N}{M} \sum_{y_i \sim \mathcal{B}} \log p(y_i \,|\, \boldsymbol{\theta}) \,. \tag{32}$$

where $M$ is the number of points in the minibatch. At the cost of increase "randomness", we can use Eq. (29) to compute the gradients of the ELBO with the minibatch formulation in Eq. (32). Stochastic optimization, e.g. any version of SGD, will converge to a local optimum provided with a decreasing learning rate and sufficient gradient updates [85].

