# OpenReview forum: "On permutation symmetries in Bayesian neural network posteriors: a variational perspective"
_NeurIPS.cc/2023/Conference — NeurIPS 2023 poster_

### Official Review · Reviewer_hffp · 2023-07-02

**Soundness:** 3 good
**Presentation:** 3 good
**Contribution:** 2 fair
**Rating:** 7
**Confidence:** 4

**Summary:**

This paper considers mode-connectivity in the context of Bayesian neural networks.  It shows roughly what we'd expect, based on the results of Entezari et al. (2022).

**Strengths:**

It shows roughly what we'd expect, based on the results of Entezari et al. (2022).

**Weaknesses:**

That's my main issue with the paper.  Bayesian neural networks really aren't that different from neural networks.  So we would definitely expect the results of Entenzari et al. (2022) to apply in the BNN context.  I can't see any interesting contributions on top of that, so I can't recommend acceptance.  I would recommend submission to a more specialised venue (e.g. AABI or UAI).

Other points:
* An important reference for permutations: Aitchison, Laurence, Adam Yang, and Sebastian W. Ober. "Deep kernel processes." International Conference on Machine Learning. PMLR, 2021.
* Legend for Fig. 8 is _way_ too small.  In general the Figures feel a bit crammed in.

**Questions:**

N/A

---

> ### Author Rebuttal · Authors · 2023-08-08
>
> We appreciate the reviewer's feedback, but we respectfully disagree with their assessment of our paper.
> Below is our best attempt at addressing the reviewers' concerns:
>
> * **The paper shows expected results**: The reviewer expresses their main issue with the paper, stating that Bayesian neural networks are not significantly different from neural networks, and therefore, the results are expected based on Entezari et al. (2022). However, while Bayesian neural networks share some similarities with neural networks, their probabilistic nature and approximate inference techniques introduce complexities not present in standard neural networks. As such, zero-barrier connectivity in the context of Bayesian neural networks is a non-trivial extension and a novel contribution to the field. Our work demonstrates that connectivity can indeed be achieved with approximate Bayesian neural networks, offering insights into their behavior and potential applications in uncertainty quantification.
>
> * **Lack of interesting contributions**: This point is very much linked to the previous one. The reviewer states that they cannot see any interesting contributions beyond the prior work. We strongly disagree with this assessment. Our paper introduces the notion of linear connectivity in Bayesian neural networks, providing a formalism as well a novel perspective on understanding their posterior landscapes. Additionally, we propose a tractable algorithm based on the Linear Assignment Problem to connect approximate solutions efficiently. These contributions offer valuable insights and open up new avenues for exploration in Bayesian neural networks.
>
> * **Recommendation for Specialized Venues**: The reviewer suggests that our work would be more suitable for a specialized venue, such as AABI or UAI. While we appreciate the suggestion, we believe that our work addresses an important question in the field of deep learning and Bayesian neural networks, making it relevant to a broader audience. Mode connectivity is a topic of significant interest, and demonstrating its applicability in the context of BNNs, along with the proposed algorithm, contributes to the understanding of optimization landscapes and uncertainty modeling in deep learning. We respectfully maintain that our paper is well-suited for consideration in the current venue.
>
> For the remaining points:
>
> * **Additional Reference**: We appreciate the reference provided by the reviewer on permutations. We will carefully consider it and include it in our revised manuscript.
>
> * **Figures and Presentation**: See the general comment. In the revised version, we will ensure that the figures are appropriately sized and presented for better clarity and readability.

---

### Official Review · Reviewer_T6Qt · 2023-07-02

**Soundness:** 4 excellent
**Presentation:** 3 good
**Contribution:** 3 good
**Rating:** 7
**Confidence:** 5

**Summary:**

The authors conjecture that after accounting for permutation symmetries in overparametrized neural networks that lead to same functional behavior, the low-loss solutions are linearly connected. In the context of Bayesian neural networks with variational inference, the authors use this conjecture to propose a simple matching algorithm based on Linear Assignment Problem, to find an equivalent variational distribution but attains linear connectivity with another solution. The experiments demonstrate that it is possible to find such linearly connected no-barrier regions.

**Strengths:**

- The authors take the ideas of mode connectivity in the deep learning literature, and try to find an equivalent formulation at the level of distributions over parameters as in Bayesian neural networks. It is a very interesting concept.
- The concept becomes even more effective by a successful demonstration of an algorithm that finds such a low-loss barrier. However, I do want to say that the existence is not surprising given that the deep learning literature already demonstrates the existence and the ELBO is just optimizing for "another" marginalized likelihood instead of an un-marginalized likelihood typically used in DL.


**Weaknesses:**

- I believe that the title of the paper should qualify variational BNNs instead of just BNNs. Approximate VI remains practically distinctive in properties from other approximate Bayesian inference methods like MCMC/HMC. It also helps contextualize the scope for the reader, and something that the authors readily align with in the discussion as well.
- Definition 3 about barrier loss seems to be connected to $\mathcal{L}$ instead of $\mathcal{L}_{\mathrm{ELBO}}$. We are interested in the loss computed by the marginalization of the model parameters under the approximate posterior. The authors choose this to be computed on the test data, which seems like jumping ahead to information that should not be used by the modeler. See also Question 1. Figure 3 shows such an algorithm does not benefit train landscape at all. Or did I misunderstand?
- In Line 159, the authors claim to argue that Wasserstein distance is better. Is the argument that covariance information is lost? I think the readers would appreciate a tiny note on using KL-divergence in the appendix and some results showing failure or becoming non-informative. See also Question 3.
- A key missing feature in the experiments is that the identification of low barrier region is not followed by the construction of posterior predictive distribution to compute the generalization error. Only likelihoods are reported, and I am wondering if the benefits of Bayesian model averaging shine as well as or better than reported in earlier works. Can the authors report those numbers? At least a basic SGD, VI, and the Aligned VI proposed in this work.
- The design choice of skipping data augmentation severely restricts the applicability of these results, especially with vision problems where data augmentation is still commonly used. I do, however, empathize with the authors on this one and do not consider this to be a big limitation to discount the contributions, since the rest of the community suffers with this limitation too.

### Minor
- Please use `\citet` instead of `\citep` when referring directly to papers. For instance, in Line 134.
- A very small description of the original LAP problem (introduced in Linear 167) in the appendix would be much appreciated.


**Questions:**

1. The posterior predictive likelihood computed on test points in Eq. (5) is also used to define the functional loss barrier in Definition 3. Don't we want the loss barrier to be defined on training points?
	- Is this simply a loose usage of the definition, or did the authors specifically imply using test points to construct barrier loss? It looks like from Figure 3 that all kinds of points are used.
	- I think it is completely fine to check using the test points as a diagnostic for correlation between behavior at test and train time.
	- But then, the matching doesn't seem to impact train landscape at all. Can the authors comment on this, since if I understand this correctly, this only remains a diagnostic method and not a method to actually generate samples from the posterior.
2. Did the authors try interpolating with a convex mixture of two distributions? There's no need to report results for this, but generally curious if such an interpolation provided some reasonable results since the authors only claim that this choice is trivial and non-informative.
3. What would using a KL-divergence instead of Eq. (16) do in practice? Would the distances be always too large to meaningfully distinguish for the LAP problem?
4. Have the authors considered accounting for such functional symmetries during the training itself, instead of a post-hoc matching procedure?

**Limitations:**

Yes. Also see, weaknesses and questions.

---

> ### Author Rebuttal · Authors · 2023-08-08
>
> We appreciate the detailed review and valuable feedback provided by the reviewer. We have carefully considered each point raised and addressed them below.
>
> * **Scope of the claim**: The reviewer suggests that the title should qualify "variational BNNs" instead of "BNNs" to better contextualize the scope of the paper. Given that this point was raised by Reviewer 2rWU as well, we will be changing  Conjecture 1 to better reflect the focus on the variational approximation.
>
> * **Definition of Barrier Loss and Use of Test Points**: The reviewer correctly points out that Definition 3, which defines the barrier, is connected to the posterior predictive likelihood computed on test points, which may seem to incorporate information not available at train time. We acknowledge this concern and apologize for any confusion. In practice, the barrier loss can be computed on both training and testing points. Indeed, all plots report both train and test barriers. We will clarify this in the revised manuscript (line 102 will be changed from "$\{x_\star, y_\star\}$ are respectively the test point and its corresponding label" to  "$\{x_\star, y_\star\}$ are respectively the input point under evaluation and its corresponding label". We also agree with the reviewer's observation that the matching procedure may not significantly impact the train landscape. The primary aim of our method is to establish linear connectivity between approximate Bayesian solutions rather than improving the train landscape.
>
> * **Use of Wasserstein Distance**: The reviewer requests further clarification on why we argue that the Wasserstein distance is better than using KL-divergence. For this we refer the reviewer to the Appendix, where we show that the KL-divergence simply reduces to a distance between means, which disregards any information regarding covariances. In the appendix, we also show a simple motivating example of such failure.
>
> * **Data augmentation**: Skipping data augmentation in our training setup was purely a choice of convenience. We know that with data augmentation we need to be careful once moving to Bayesian inference for a multiple of reasons (e.g., cold posterior effect [92], re-weighting of the likelihood due to the increase of the effective sample size [68]). For this reason, we didn't want to "pollute" the results with spurious effects coming from DA, rather than the phenomenon under analysis. Nonetheless, we want also to emphasize that nowhere during the development of the method we make an assumption on DA, which makes our method still applicable in both cases. During this rebuttal week, we were able to run some comparisons with data augmentation for the ResNet20 models. Figure 8 in the rebuttal PDF summarizes this experiment: we see that in both cases (with and without DA), we are still able to recover similar low barrier solutions in terms of likelihood (and accuracy---not reported for space reasons) when following our proposal to align the distributions. As discussed in the general comments, we are planning tp add an additional paragraph in Sec. 5 to better comment on the effect of DA.
>
> * **Benefits of BMA**: In terms of generalization on accuracy, indeed BMA is  beneficial w.r.t. point estimates. See the table below, where we report the accuracy of the two models, as well as the interpolated one $\tau=0.5$.
>
> **ResNet/CIFAR10 (Accuracy)**
>
> |                | Model 0 | Model interpolated | Model 1 |
> | ---------------|---------|--------------------|---------|
> |             VI | 0.8580  |       0.1025       | 0.8556  |
> | **VI aligned** | 0.8580  |       0.7413       | 0.8556  |
> |            SGD | 0.8558  |       0.1432       | 0.8516  |
>
> **MLP/CIFAR10 (Accuracy)**
>
> |                | Model 0 | Model interpolated | Model 1 |
> | ---------------|---------|--------------------|---------|
> |             VI | 0.5718  |       0.2589       | 0.5734  |
> | **VI aligned** | 0.5718  |       0.5647       | 0.5734  |
> |            SGD | 0.5546  |       0.2500       | 0.5545  |
>
> These numbers are in line with previous work in the literature.
> We also report the reference comparison with the likelihood below
>
> **ResNet/CIFAR10 (Likelihood)**
>
> |                |   Model 0 | Model interpolated |  Model  1 |
> | ---------------| ----------|--------------------| ----------|
> |             VI | -0.417142 |      -2.32982      | -0.427943 |
> | **VI aligned** | -0.417142 |      -0.71424      | -0.427599 |
> |            SGD | -0.702219 |      -2.43758      | -0.731933 |
>
> * **Mixture of distributions**: as we discussed for Reviewer 2rWU, we argue that a mixture is not informative for studying the geometry of the posterior. To visualize this argument, please check Figure 1 in the rebuttal PDF. With mixtures we are essentially re-weighting the two (good) solutions, without "transport" of distribution mass between the two extremes. As we said in the general comment, this means that if we look at the barrier, we don't see any not because they don't exist, but simply because we are interpolating in such a way that prevents us from exploring the geometry of the posterior.
>
> * **KL divergence**: Thanks for the question. In the appendix we actually show how using the KL divergence in practice falls back to a distance between means (without accounting for the actual shape of the distributions). We also have a simple visualization where this fails in practice, i.e. the LAP with the KL objective fails to recover a clear symmetry.
>
> * **Accounting for Functional Symmetries during Training**: While the idea of accounting for functional symmetries during training is intriguing, it presents a challenging optimization problem. We have not explored this approach in the current paper, but it could be an interesting direction for future research. We will mention this possibility in the discussion section to encourage further investigations.

---

### Official Review · Reviewer_bvHV · 2023-07-06

**Soundness:** 3 good
**Presentation:** 3 good
**Contribution:** 2 fair
**Rating:** 6
**Confidence:** 4

**Summary:**

The authors extend the recent idea of linear mode connectivity up to permutation symmetry to the setting of Bayesian neural networks. They demonstrate that two different variational approximations to the Bayes posterior enjoy mode connectivity along the Wasserstein geodesic of one distribution, and a suitably permuted version of the other. Such a permutation is discovered by replacing the L2 distance in previous work with the more suitable Wasserstein distance, and similarly,  the objective can then be relaxed to a layer-wise linear assignment problem, leading to a tractable coordinate descent algorithm. The authors verify their results numerically, showing that two approximate solutions can indeed be connected for modern networks such as ResNet20 on CIFAR10.

**Strengths:**

1. Studying mode connectivity for approximate Bayesian inference is a natural follow-up question to previous work, while at the same time requiring non-trivial extensions such as the Wasserstein geodesic and Wasserstein distance. It is very nice and somewhat surprising that the resulting objective (which is seemingly involved) can be relaxed in a very similar spirit, leading to a tractable problem.
2. The experimental setup is quite carefully created and a lot of ablations for different 	parameters such as the prior variance, the temperature and the width of the network are performed, giving a very complete picture.

**Weaknesses:**

1. The proposed algorithm and the setup seem to heavily rely on the specific approximation method, namely variational inference with a Gaussian distribution of diagonal covariance as the variational family. Even if the diagonal covariance assumption is relaxed, it is not obvious to me how one can guarantee tractability, let alone moving to multi-modal approximations such as MCMC where not even the Wasserstein geodesic or distance are known in closed form. This is somewhat unsatisfying, since the power of BNNs and the Bayesian posterior in general only really starts to unfold once multiple modes are leveraged. It is thus somewhat unclear how much a unimodal approximation to the Bayes posterior really captures and how much it is really different from a simple point estimate.
2. In general, the problem seems to get less and less interesting, the more precise the approximation to the Bayes posterior becomes. This is simply because the Bayes posterior would incorporate all possible modes (given the prior gives them some mass) and hence there is no other posterior to align with. This is different from the SGD setup, where multiple modes always remain a problem, precisely due to the point-wise nature of the algorithm. The authors are very careful in the main text and always refer to approximate Bayesian inference, but I think it would be helpful to clarify this discrepancy.
3. I understand why the authors use the log-likelihood as a metric to evaluate connectivity, given that it is a proper scoring rule, but it is also very difficult to interpret how meaningful a decrease in likelihood is for practice. This is in contrast to test accuracy, where we have a better understanding of the scale. It would be helpful if the authors could provide the same plots for test accuracy instead of log-likelihood. That way it would also be easier to assess how meaningful the barrier in Fig. 3 for ResNet20 actually is. It would also be helpful to reproduce the same plots as in Fig. 3 without adding the non-aligned connectivity score since this massively increases the scale. That way it is actually difficult to tell how connected the aligned solutions are. They are obviously way more connected than the baseline, which is nice but it would be better to see it in more detail.

**Questions:**

1. Is data augmentation employed for the experiments involving the cold posterior effect? Data augmentation has been observed to be the main driver of the CPE [1, 2, 3] and hence using it would probably lead to a more visible effect. It has also been recently justified that tempering is a principled way to use data augmentation in Bayesian frameworks [4], so it would not affect the validity of the approximation.
2. Are there gains in terms of test accuracy of the (tempered) approximate Bayesian posteriors versus a standard SGD baseline?


[1] What are Bayesian Neural Network Posteriors Really Like?,
Izmailov et al.

[2] Data augmentation in Bayesian neural networks and the cold posterior effect,
Nabarro et al.

[3] Disentangling the Roles of Curation, Data-Augmentation and the Prior in the Cold Posterior Effect,
Noci et al.

[4] How Tempering Fixes Data Augmentation in Bayesian Neural Networks,
Bachmann et al.

**Limitations:**

The authors have addressed the limitations of their work.

---

> ### Author Rebuttal · Authors · 2023-08-08
>
> We thank the reviewer for their constructive feedback and insightful comments.
>
> * **Tractability of Approximation Methods**: The reviewer raises a valid concern regarding the reliance of our proposed algorithm on specific approximation methods, such as variational inference with a Gaussian distribution of diagonal covariance. We agree that the power of Bayesian neural networks lies in capturing multiple modes, and unimodal approximations may limit their expressiveness. While our method is tractable for the chosen approximation, we acknowledge that extending it to more complex multi-modal approximations like MCMC could be challenging but not intractable (e.g. the Wasserstein geodesics can be approximated with the Sinkhorn algorithm [17 in paper]).
>
> * **Precision of Approximations and Interpretability**: The reviewer correctly notes that as the approximation to the Bayes posterior becomes more precise, the problem of  connectivity becomes less interesting, as the Bayes posterior would naturally incorporate all possible modes. While we agree with the reviewer, we also need to acknowledge that despite the best efforts from the community, the BNN posterior is still elusive: even carefully tuned MCMC methods cannot fully characterize the full extent of its nature. And they still needs orders of magnitude more of gradient evaluations. We believe that this characterization of permutation symmetries will have value for SG-MCMC methods as well.
>
> * **Comparison with SGD**: Regarding the comparison with SGD, it's still up to debate whether SGD solutions are actual global minimizers (the same Ainsworth's paper shows that we can find better solutions than the ones found by pure SGD, supporting the hypothesis that SGD solutions are not global). In this regard, we can view SGD as an approximation method of the exact minimization problem $\min_\theta\ell(\theta)$, for some loss function $\ell(\cdot)$, in the same way in which we can view approximate Bayesian inference (including sampling) as an optimization problem over space of probability measures w.r.t. the true posterior (although it's clear that the nature of the symmetries are not equivalent for the two setting).
>
> * **Evaluation Metric**: We appreciate the reviewer's suggestion regarding using test accuracy as an additional metric for evaluating mode connectivity. Test accuracy is indeed more interpretable in practice, and we agree that including it in the evaluation would provide valuable insights. In response to this suggestion, we will incorporate plots of test accuracy alongside log-likelihood in the revised manuscript, offering a more comprehensive assessment of the model's performance and connectivity. In the meanwhile, please check in the rebuttal PDF the replication of Figure 3 with the accuracy, along side with the requested zoom.
>
> * **Data Augmentation and Cold Posterior Effect**: We appreciate the reviewer pointing out the potential impact of data augmentation on the cold posterior effect (CPE). No, data augmentation is never employed in our experiments (see general comments). Having said that, during this rebuttal we were able to run a comparison on the effect of the data augmentation alone (without tempered posteriors). In Figure 8 of the rebuttal PDF, we can see that the behavior for both with and without DA is very similar. We will conduct additional experiments with data augmentation and temperature scaling to provide a more comprehensive analysis of its influence on our results. Due to the limited time we were unable to have the comparison DA+temperature, but it will be added for the next version.
>
> * **Test Accuracy Gains**: While analyzing the effect of tempered posterior is not our primary scope in our work, we report the results in the table below for ResNet20.
>
> | Method      | Accuracy |
> | ----------- | -------- |
> | Tempered VI | 0.8281   |
> | VI          | 0.8580   |
> | SGD         | 0.8546   |
>
> Because we are using a clean likelihood (no batch statistics, and no DA) the results with VI are slightly worse but in line with the literature (see Appendix K4 in [92 from paper])

---

> > ### Comment · Reviewer_bvHV · 2023-08-14
> > **Response to Rebuttal**
> >
> > I thank the authors for running additional experiments in such a short amount of time!
> >
> > **Tractability:** I agree that the Sinkhorn algorithm could be used to approximate the Wasserstein distance, but it would be tricky to reduce the permutation problem to anything tractable in this case, right? I might be missing something.
> >
> > **Precision of Approximation and Interpretability:** I agree that the exact posterior remains elusive and understanding approximate methods is thus very valuable. I simply remain unsure whether permutations really say anything fundamental about approximate posteriors. While of course SGD is also not an exact minimizer of its objective, even if it were, the permutation problem still remains! Moreover, here the permutation symmetry really reveals something fundamental about the problem.
> >
> > **Evaluation Metric:** Thank you for adding this! The precise definition of linear connectivity is always an issue in this line of work. What do the authors consider as connected here? Strictly speaking, none of the results are linearly-connected as the values do worsen, as evident now in the zoomed plot. I'm aware that similar results might also have passed as being "connected" in the literature, so I don't want to impose a new standard here, but it would be great if the authors could at least compare their connectivity values (for accuracy) with [1]. Especially the ResNet seems to worsen by almost 10% on the training data (removing the baseline here too would be helpful), which seems somewhat drastic. I hope the authors can clarify this.
> >
> > **Data augmentation:** Thank you for clarifying this! It's nice that data augmentation does not affect results to drastically.
> >
> > [1] Git Re-Basin: Merging Models modulo Permutation Symmetries, Ainsworth et al

---

> > > ### Author Response · Authors · 2023-08-15
> > >
> > > We thank the reviewer for these additional comments. Here's our reply:
> > >
> > > * **Tractability**: We agree with the reviewer on this point. Extending this to sample-based inference is not going to be trivial nor easy, hence our comment on leaving this as future work.
> > >
> > > * **Precision of Approximation and Interpretability**: We can give the reviewer one possible application where such permutation analysis can be beneficial for approximate inference: convergence analysis of SG-MCMC. We know by experience that classic convergence statistics (like R-Hat) are not robust to assess convergence behavior of MCMC chains in large models. In Figure 2 in [1] we see that despite unrealistic compute availability, the R-Hat statistics severely underestimates the convergences of the chains, due to the exploration by HMC of permutation equivalent modes. While it is possible to compute such statistic in function-space, technically this is not a standard practice, since the R-Hat becomes also a function of the inputs, rather than just being a property of the chains/samples. On the other hand, by accounting for permutation symmetries we could derive more appropriate convergence statistics for MCMC methods to be used in this context.
> > >
> > > * **Evaluation Metric**: If we look at Figure 9 in [2] we see that for the MLP the results are actually very similar with ours (if not marginally worse, see CIFAR10/MLP). The biggest difference is indeed with ResNet20, where the only difference is the normalization layer (LayerNorm for [2] and FRN for us). We speculate this being the cause. Nonetheless, we want to highlight how our alignment method decreases the (accuracy) barrier by 85.89% on the train set and by 87.55% on the test set.
> > >
> > >
> > > [1] Pavel et al. What Are Bayesian Neural Network Posteriors Really Like?
> > >
> > > [2] Ainsworth et al. Git Re-Basin: Merging Models modulo Permutation Symmetries.

---

> > > > ### Comment · Reviewer_bvHV · 2023-08-17
> > > >
> > > > **Interpretability:** I'm not sure if I completely understand. Wouldn't the algorithm from *Ainsworth et al.* suffice to perform a permutation check between two MCMC samples (as far as I know, no Gaussian is built around MCMC samples)?
> > > >
> > > > **Evaluation Metric:** Thanks for the comparisons! FRN vs LayerNorm could be the reason that the ResNet20 is less connected but of course it could also be the novel alignment procedure proposed in this work. The only way to know for sure is to either check the novel procedure with the same LayerNorm-based ResNet20 or compare against other connectivity results for the FRN ResNet20 in the literature.

---

> > > > > ### Author Response · Authors · 2023-08-17
> > > > >
> > > > > **Interpretability**: with MCMC, we would not be interested in single MCMC samples but rather on populations of samples (e.g. comparing different chains). In this sense, Ainsworth et al. is not directly applicable here.
> > > > >
> > > > > **Evaluation Metric**: this was exactly what we were running this past few days. It's not clear whether we can add links to external services but if you are interested here ([https://imgur.com/a/0iHJxiY](https://imgur.com/a/0iHJxiY)) you'll find a comparison between FRN and LN (hopefully we are not braking any rules, here). If you prefer not to follow the link, the gist of the figure is confirming that LN is performing better than FRN, further reducing the barrier by ~35%.
> > > > >
> > > > > | Normalization     | Barrier Accuracy    | Barrier Likelihood   |
> > > > > | ----------------- | ------------------- | -------------------- |
> > > > > | FRN (with align.) | 0.163350            | 0.519373             |
> > > > > | LN (with align.)  | 0.105150 (-35.629%) | 0.336379 (-35.2336%) |

---

> > > > > > ### Comment · Reviewer_bvHV · 2023-08-17
> > > > > >
> > > > > > **Interpretability:** I see, so that's kind of the multi-modal case though that's not doable now. But this work might serve as a first step towards this. At least the mathematical formulation might be helpful here.
> > > > > >
> > > > > > **Evaluation Metric:** Thanks a lot for running this! Are those test scores? And for accuracy, is the barrier for LN 10%? What was the value for *Ainsworth et al.*?

---

> > > > > > > ### Author Response · Authors · 2023-08-17
> > > > > > >
> > > > > > > **Evaluation Metric**: Yes, those are the test score. Note that the width of the model is different: for sake of compute time, we are showing the x8, while they only have the x32 (sorry for the missing information above).  Also Ainsworth et al. doesn't report this number

---

> > > > > > > > ### Comment · Reviewer_bvHV · 2023-08-19
> > > > > > > >
> > > > > > > > Thank you for all the clarifications. They mostly address my concerns and I have raised my score. I think for the final version of this paper, it would be great to have a 1-1 correspondence with the setup in *Ainsworth et al.* so that connectivity values can actually be compared. I still do not feel completely certain whether a 10% drop qualifies for enough connectivity.

---

### Official Review · Reviewer_Kavz · 2023-07-06

**Soundness:** 4 excellent
**Presentation:** 4 excellent
**Contribution:** 3 good
**Rating:** 7
**Confidence:** 2

**Summary:**

This work permutes together the distributions of bayesian neural network (BNN) parameters in the context of variational inference (VI) so that they are linearly connected. This is done by adapting recent work on permuting together SGD solutions to be linearly connected, by optimizing for similarity of means and variances from VI in the permutation assignment objective, instead of similarity of parameters (as in Ainsworth et al. 2022). The problem setup and permutation objective are described and derived in detail. Experiments show that this method is comparable in loss barrier to directly permuted SGD solutions.

**Strengths:**

Bringing permutation alignment into a probabilistic setting is novel, and the formalization of the methodology is satisfying. The paper is well-organized, and a number of variations (variance of prior, temperature) are considered in the experiments. Despite its focus on BNNs, this work also has relevant implications for the SGD solution setting, since it would be nice to be able to extend linear mode connectivity from individual solutions to families of SGD solutions whose parameters are stochastic (e.g. due to randomness in initialization/training).

**Weaknesses:**

In both the temperature and prior variance experiments, a comparison of barriers at the limit of 0 variance/temperature (corresponding to direct alignment of the MAP solution) for CIFAR-10 would be interesting, as it would put the observed difference between VI alignment and SGD alignment in figure 8 into context. Specifically, it would be nice to see figures 6 and 7 replicated on CIFAR-scale networks, and a reference line  included in each figure to indicate the barrier achieved by SGD alignment.

From a presentation standpoint, figures 4-8 are arranged strangely (5 is out of order) and too small to read clearly. In particular, the text and lines should be larger, some of the margins smaller, and more distinct colors used. Figures 1-3 are less important from a readability standpoint, but may also benefit from the same adjustments.

**Questions:**

Given that barrier is higher when aligning VI distributions versus aligning the MAP parameters (akin to directly aligning SGD solutions), what are the potential benefits of optimizing alignment from a distributional standpoint? If the goal is to minimize barrier, why not simply align the MAP? Some motivation would be helpful here.

**Limitations:**

The main limitation (which is briefly discussed in section 7, but not strongly emphasized throughout) is that the assumptions on the posterior distribution are too rigid: namely that parameters are independent Gaussians. The authors conjecture (line 239) that this may be the cause of reduced performance relative to Ainsworth et al. (2023). Given the derivation in (16) already admits arbitrary covariances, it seems very interesting to consider the case where the covariance of the posterior is non-diagonal (lines 275-276). This could also lead to significant innovations over the existing alignment algorithm of Ainsworth et al. (2023), which is largely unchanged in this work.

---

> ### Author Rebuttal · Authors · 2023-08-08
>
> We thank the reviewer for insightful comments and discussion points.
>
> * **Temperature and prior for ResNet20/CIFAR**: Thanks for the suggestion, we will take this into consideration. During the limited time span of this rebuttal, we were able to run the ResNet comparison with different prior variances and different temperatures (see Figure 7 in the rebuttal PDF). Modulo the fact that finding zero-barrier is more difficult on ResNet (vs MLPs), the comments for prior variances in the paper generally applies for this experiment as well: "without alignment we see that naive VI solutions are easier to interpolate with lower barrier when the prior is more diffused. At the same time, we see that higher variances produce bigger gaps between train barriers and test barriers. We speculate that this is due to overfitting happening with more relaxed priors, which makes low-barrier (but low-likelihood) solutions easier to find". Similar for the temperature comparison, for which "we see that barriers for cold posteriors with alignment are marginally closer to zero than for warm posteriors". To conclude, in both cases the results are consistent with the comments in the paper, but we will add these additional experiments to strengthen the discussion. Thanks for the suggestion!
> * **Presentation**: Thanks, this is already taken into consideration for the next version of the manuscript (see comment to Reviewer 2rWU). We will make sure to improve the readability of the plots.
> * **Aligning using MAP**: Thanks for the question. Indeed, we could align the MAP solutions and take the permutation symmetry found. Still, we don't think this will work in practice for few reasons: (i) independent models do not share the same permutation matrices, (ii) if we look only at the means, the new results requested by Reviewer Kqgj show that this is indeed not the best objective for alignment.
> * **Limitation on the posterior**: Yes, indeed extensions to non-diagonal covariances are possible. We haven't investigated this possibility yet for a couple of reasons: (i) full covariances with variational inference are intractable for these large models, resorting in various possible approximations to computationally scale to these networks (low-rank, structured covariance, etc.); (ii) with non-diagonal covariances the steps to go from Eq. (16) to the formulation in Eq. (19) would not be possible. A way to approach this could be by learning the permutation $P$ using a new ELBO together with a straight-through estimator:
>
> $$
> \max_{\widetilde{q_1}} \mathcal L_{elbo}\left(\frac 1 2 (Id+T_{q_0}^{\widetilde{P}\\#\widetilde q_1})\\# q_0 \right) \quad \text{where} \quad \widetilde{P} = \arg\min_P \mathcal W(P_\\# q_1, q_0)
> $$
>
> To make it tractable, the r.h.s. needs to approximate the full covariance with its diagonal components, while the l.h.s. should be computed with its full covariance. We believe that such discussion requires some careful analysis, which was beyond the scope of this work.

---

> > ### Comment · Reviewer_Kavz · 2023-08-16
> >
> > Thank you for the additional results and the clarification regarding tractable posteriors. Given the new results show the advantage of including covariance in the alignment objective, I will raise my score.

---

### Official Review · Reviewer_2rWU · 2023-07-19

**Soundness:** 3 good
**Presentation:** 3 good
**Contribution:** 3 good
**Rating:** 6
**Confidence:** 4

**Summary:**

The authors study the geometry of SGD-trained Gaussian mean-field variational approximations to the posteriors of Bayesian neural networks (BNN). In large part, the authors propose extensions of the method and analysis of Ainsworth et al. [1] from MAP-estimated neural networks to BNNs. Notably, the authors informally conjecture that a permutation symmetry exists in the solutions that approximate Bayesian methods can find, similar to the permutation symmetry of MAP solutions SGD finds. To test this hypothesis, the authors first use optimal transport theory to extend the linear interpolation and the loss barrier framework of [1] to the variational posterior approximation setting. Then, they show that they can align individual mean-field Gaussian weight posteriors by approximately solving a bilinear assignment problem using an algorithm analogous to the one proposed by [1] for MAP solutions. The authors demonstrate empirically that zero-barrier interpolations exist for permutation-aligned variational posteriors of non-trivial architectures (ResNets) trained on non-trivial datasets (MNIST, FashionMNIST and CIFAR-10).

## References

[1] Ainsworth, S. K., Hayase, J., & Srinivasa, S. (2022). Git re-basin: Merging models modulo permutation symmetries. arXiv preprint arXiv:2209.04836.

**Strengths:**

Understanding the properties of approximate solutions to the posteriors of BNNs is one of the central challenges of Bayesian deep learning; hence extending the method and analysis of Ainsworth et al. to the variational BNN setting is an important step towards this goal. The proposed extension of the interpolation framework and the alignment procedure using optimal transport theory seems natural, and the experimental methodology is sound and reasonably thorough. Besides the experiments analogous to the ones of Ainsworth et al., the authors also investigate the effect of prior variance and the cold posterior effect, the limit of which is the MAP solution. The paper is mostly well-written and easy to follow.

**Weaknesses:**

While I did not find any significant weaknesses in the work, there are a couple of points that, if improved/clarified, could significantly strengthen the paper.

### Conjecture 1

First, I found Conjecture 1 too broad and imprecise, even compared to the conjecture given by Entezari et al. [1], mainly for two reasons. The conjecture states that:

"Solutions of approximate Bayesian inference for neural networks are linearly connected after accounting for functionally transparent permutations."

First, I think including the class of approximate Bayesian methods is too broad because it covers variational inference (VI), Markov chain Monte Carlo (MCMC), and the Laplace approximation, which all yield significantly different solutions. Hence, since the authors only study mean-field VI in the paper, I think it would be better to restrict the conjecture to this case only and potentially extend the conjecture to the other methods if future work supports it with some empirical evidence.

Related to the first point, it is thus unclear if there is a universal notion of "linear connectedness" for the approximate solutions I mention above. Furthermore, as both Entezari et al. [1] and Ainsworth et al. [2] pointed out, the modes' linear connectedness appears to be a particular feature of SGD. Hence, I would suggest that the authors rephrase their conjecture to something like

"SGD-based solutions of mean-field variational Bayesian inference for neural networks are linearly connected after accounting for functionally transparent permutations."

Thus, this is also a more direct generalization of the conjecture given in [1]. What do the authors think?

### Connection between the interpolation and the alignment process
The authors define the interpolation of two variational posteriors as well as the alignment procedure using optimal transport (OT) theory. However, as far as I can tell, these two things are not obviously connected. While I am not claiming that the authors' choice to use OT for these definitions is unnatural, it currently seems more of a choice of convenience than one dictated by theory. In particular, it seems that the authors chose this precisely because the alignment procedure reduces to a bilinear assignment problem, and the interpolating density is also Gaussian. Could the authors clarify whether there is some deeper theory that would justify the authors' choices? Does the alignment objective in eq (20) follow somehow from the interpolation method in eq (6)?

Relatedly, the authors state the following on L116:

"While we could interpolate using a mixture of the two solutions, we argue that this choice is trivial and does not fully give us a picture of the underlying loss landscape."

What the authors mean by the word "trivial" here is unclear. They appear to mean that interpolating with arithmetic mixtures is an "obvious" choice, not "trivial". The properties of the mixture choice are not at all obvious to me, and it is not clear why it doesn't provide us with information about the loss landscape. Again, I am not saying that the authors' choice of using OT is wrong and the arithmetic mixtures are better or equally useful; but they need to give arguments (either theoretical or empirical) why they think it is uninteresting.

In a similar vein, I wonder if using geometric mixtures, i.e. using $q_\tau \propto q_0^{1 - \beta} \cdot q_1^{\beta}$ for $\beta \in [0, 1]$, would have similar behaviour, or if it would lead to a different alignment procedure.

### Figure 5
Given the interpolation and alignment process, is the hyperplane in Figure 5 meaningful? The hyperplane is defined using a simple linear interpolation of the weight posteriors' mean parameters, which seems to go against all the previous machinery the authors argued for earlier in the paper. Perhaps if the authors want to include such a hyperplane, perhaps there's a definition that can be made using OT, or they could draw samples $ \theta_a \sim q_0, \theta_b \sim q_1, \theta_c \sim P_\sharp q_1$ and linearly interpolate those?

### Miscellaneous
Eq (21) has a small mistake, the argmax ranges over $i \in [0:L]$, but the objective only involves permutations with indices up to $L - 1$. Furthermore, Eq (21) could be compactified by defining P_0 = P_L = I and writing the sum using $\sum$ notation.
Line 4 in Algorithm 1 should be broken up into two or three lines, and the font size should be increased.
The font size in Figures 3-8 is too small and should be increased to match at least the font size of the captions.
The training procedure needs clarification. Did the authors use Bayes by Backprop [3] or the local reparameterization trick [4]?

## References

[1] Entezari, R., Sedghi, H., Saukh, O., & Neyshabur, B. (2021). The role of permutation invariance in linear mode connectivity of neural networks. arXiv preprint arXiv:2110.06296.

[2] Ainsworth, S. K., Hayase, J., & Srinivasa, S. (2022). Git re-basin: Merging models modulo permutation symmetries. arXiv preprint arXiv:2209.04836.

[3] Blundell, C., Cornebise, J., Kavukcuoglu, K., & Wierstra, D. (2015, June). Weight uncertainty in neural network. In International conference on machine learning (pp. 1613-1622). PMLR.

[4] Kingma, D. P., Salimans, T., & Welling, M. (2015). Variational dropout and the local reparameterization trick. Advances in neural information processing systems, 28.

**Questions:**



**Limitations:**

---

> ### Author Rebuttal · Authors · 2023-08-08
>
> We thank the reviewer for his/her interesting comments. Below we reply inline to the reviewer's questions:
>
> * **Strength of conjecture 1**: We agree with you on this. We decided to have a broad conjecture to leave room for possible extensions to Laplace approximation (easier) and SG-MCMC methods (definitely more challenging). Since this point was raise by Reviewer T6Qt as well, we think it's appropriate to rephrase it to limit the context under analysis. Thanks for the suggestion.
> * **Connection between interpolation and alignment**: That is correct: technically Eq (20) and Eq (6) are independent and indeed other choices can be made (interpolation using the mixtures and alignment the KL divergence). For the interpolation, the choice of the Wasserstein geodesic was dictated by the geometry of the interpolation paths between the two distributions (akin to linear interpolation for the Euclidean metric, more on this in the next point). Additionally, the use of Wasserstein distance for the alignment conveniently allows us to re-interpret the problem as an assignment problem, for which efficient routines are available. In the appendix, we briefly show why we think the KL distribution is not the best objective we can use to align distributions. Indeed, other choices can be made (see new experiment run for Reviewer Kqgj).
> * **"Mixtures are trivial"**: we realized that "trivial" was a poor choice of word in this context. What we meant by that is that the mixture of distribution is not sufficient to capture the underlying complex geometry of the posterior. To aid this conversation, please have a look at the Figure 1 in the PDF. Here, we plot the test likelihood with two interpolation strategies: OT vs mixture (both without any alignment). With mixtures, we see that the likelihood is pretty much constant during the interpolation, but this is very miss-leading: we don't see barriers not because they don't exist, but because the mixture simply re-weights the distributions, without continuously transporting mass in the parameter space.
> We will make sure to reword this passage more appropriately.
>
> * **Figure 5**: Thanks for the suggestion. Indeed, this might not be the best way to visualize the underlying structure. Nonetheless, we believe that the visualization has still value, as it shows the different connectivity property between the aligned and the not-aligned solutions. Having said that, we took the Reviewer's opinion into account and we prepared another visualization (see Figure 2 and 3 in the rebuttal PDF). In this case we sampled from the two solutions, as well as from the interpolated one (for $\tau=0.5$), with and without alignment. We then use these three sample as support to build the projecting hyperplane. For sake of visualization, these three samples once projected are not on a line, but in the original space they are. With alignment, we see that the three samples all lie in a relatively flat region on the posterior, in contrast to the case of no alignment where we once more see the raising of a barrier when connecting the three samples.
>
> * **Miscellaneous**: Thanks for spotting the mistake in Eq (21), you are right. Font size and figure placement were known issues, unfortunately we didn't find other arrangements that would fit in 9 pages. Eventually, if allowed to add an additional page for the camera-ready, we will make sure to fix this. For the training procedure, we refer back to the details in the Appendix (we are indeed using the reparameterization trick, not the local)

---

> > ### Comment · Reviewer_2rWU · 2023-08-11
> > **Response to the Authors**
> >
> > I thank the authors for their rebuttal. After reading the other reviews and the authors' rebuttals, I maintain that the paper should be accepted and I raise my confidence score to reflect this.
> >
> > I thank the authors for providing a very nice explanation of why the mixture interpolation is unhelpful. I think they should also include Figure 1 in the appendix of the camera-ready version of the paper, as it would help readers who do not have a strong background in OT (like myself) to understand the authors' choices much easier. Though they did not address my point regarding geometric mixtures, I am now much more convinced that the OT formulation is more appropriate, as it captures the intuition of linear interpolation analogous to the point-estimate setting.
> >
> > *Regarding the disconnect between the interpolation and alignment procedure:* I think, in a certain way, the missing puzzle piece for me regarding the OT-based formulation is how SGD is connected to it. In particular, I wonder if using SGD to train the VI solution could be construed as some sort of OT procedure to transport the initial posterior guess to some optimal one. For example, is it connected to the Wasserstein gradient flow (Salim et al., 2020)? I think some form of a positive answer would conclusively put my worries to rest since this would shed light on how the VI setting generalizes the point-estimate setting. If SGD on VI was performing OT, it would be much more intuitively obvious why we should expect the OT interpolation and alignment procedures to be the right things to look at. Perhaps this could also help establish a connection between the two, though this is more of a shot in the dark on my part.
> >
> > ## References
> >
> > Salim, A., Korba, A., & Luise, G. (2020). The Wasserstein proximal gradient algorithm. Advances in Neural Information Processing Systems, 33, 12356-12366.

---

> > > ### Author Response · Authors · 2023-08-11
> > >
> > > Thanks for reply.
> > > Regarding the geometric mixture, we apologize for missing this point earlier. We are not sure how the interpolation paths will look like with a geometric mixture, especially at the extremes ($\tau=\{0,1\}$). Indeed if $q$ is Gaussian, we have
> > >
> > > $$
> > > q^\tau \propto \exp\left(-\frac \tau 2 \left(\frac{x-\mu}{\sigma}\right)^2 \right) =  \exp\left(-\frac 1 2 \left(\frac{x-\mu}{\sigma/\sqrt{\tau}}\right)^2 \right)
> > > $$
> > >
> > > which is the un-normalized PDF of a Gaussian RV with variance $\sigma^2/\tau$ (with $\tau \rightarrow 0$ we have a degenerate distribution).
> > >
> > > Regarding the connection with the Wasserstein gradient flow, an interesting reference connecting VI and OT could be [1]. In this paper, the authors show how the dynamics of the mean and (co)variance of Gaussian VI follows the gradient flow of the Kullback–Leibler (KL) divergence $KL(\cdot \| \pi)$ on a submanifold of the Wasserstein space of Gaussian distributions on $\mathcal P_2(\mathbb R^d)$ (known as Bures–Wasserstein manifold), which is equipped with the 2-Wasserstein distance.
> > > Additionally, the authors also show an SGD version for the discretization of the Bures–Wasserstein gradient flow, which might represent an interesting results to further characterize the properties of the approximate neural network posterior.
> > > Thanks again for this discussion.
> > >
> > > [1] Marc Lambert, Sinho Chewi, Francis Bach, Silvère Bonnabel, Philippe Rigollet. Variational inference via Wasserstein gradient flows. NeurIPS 2022

---

### Official Review · Reviewer_Kqgj · 2023-07-26

**Soundness:** 3 good
**Presentation:** 3 good
**Contribution:** 3 good
**Rating:** 7
**Confidence:** 4

**Summary:**

This paper paper extends linear mode connectivity modulo permutation to Bayesian neural networks posteriors. Authors do this by imposing a Wasserstein metric on the space of distributions and look at how log likelihood changes along the geodesic between two distributions obtained via approximate Bayesian inference. For experiments, authors study BNNs parametrized by multivariate Gaussians with diagonal covariance. Analogous to weight matching algorithm proposed in [1], authors propose to find permutation that minimizes Wasserstein distance between the two distributions. For Gaussian distribution with diagonal covariance, authors propose an algorithm similar to weight matching that also takes into account the variances of the Gaussian. In experiments authors show that this heuristic for finding permutation that leads to small change in  log likelihood along the geodesic.

*[1] Git Re-Basin: Merging Models modulo Permutation Symmetries. Ainsworth et al.*

**Strengths:**

- Overall a well-written paper that extends previously studied linear mode connectivity modulo permutation of neural networks to Bayesian neural networks.
- Experiments demonstrate that the heuristic proposed to compute permutations works for network architectures and datasets studied in the paper.

**Weaknesses:**

- Limitations of the proposed heuristic are not extensively discussed in the paper. For instance model architectures with batch norm typically fails without additional fixes [2].
- It’s not clear if the proposed algorithm out performs standard weight-matching / activation matching / STE estimator based approach to finding permutation discussed in [1].

*[2] REPAIR: REnormalizing Permuted Activations for Interpolation Repair. Jordan et al.*

**Questions:**

Empirically, how important is the role of adding covariance matrix in the objective when computing the permutations? For instance, using standard weight-matching in this setting just assumes that the covariance matrix is identity. it would be useful to see a table with comparison of different approaches.

**Limitations:**

This paper limits itself on studying connectivity modulo permutation for approximate Bayesian NNs posteriors found via gradient based algorithms. It’s not clear if this phenomenon holds for other approaches which should be interesting future work.

---

> ### Author Rebuttal · Authors · 2023-08-08
>
> We thank the reviewer for his/her interesting comments:
>
> * **Model architecture**: we agree that the choice of model architecture can impact the ability to find good solutions with low/zero barrier. One of this choice is the width of the neural network, for which the wider the model, the lower the barrier is. We are aware that the choice of the normalization strategy (especially batch-dependent) can affect the overall geometry of the problem (exactly how demonstrated in Fig.3 in your reference). Additionally, batch-dependent normalization layers don't have a clear Bayesian interpretation (technically, the likelihood in line 84 cannot factorize). That's why we chose to use the Filter-Response-Normalization layer, as done in other previous work [e.g. 40 in paper]. Having said that, we think it would be possible to apply the REPAIR method to our case as well, but I would expect the FRN to behave similarly to LN. For quick sanity check, we analyze the variance of activation following the instructions in [2], with the sole difference that the activations are marginalized w.r.t. samples from the posterior; for both naïve interpolation and aligned interpolation we then compute the variance ratio as discussed in Section 3.1 of [2] (we take $\tau=0.5$ as middle point between the two “endpoints”). Results for a couple of models are available in the rebuttal PDF. We can see that there seems to be a level of collapse for the various configurations, which despite being less pathological than the one shown in [2, Figure 2] and improving using the distribution aligned method, appears to be an effect present in the Bayesian settings as well. Finally, we realized that REPAIR is not cited in the current version of the paper, we will make sure to add it in the next version. Thanks for the interesting point!
>
> * **Performance w.r.t. to [1]**: Figure 8 suggests that weight-matching (which was overall performing the best among the three in [1]), can achieve slightly better performance w.r.t. distribution alignment especially for wide networks (albeit with overall lower likelihood).
>
> * **Effect of (co)variance**: please refer to the table below, where we report the barrier with different alignment objectives:  the naïve interpolation (without any alignment) and two types of alignment objective (full, which reflects the correct objective as derived in Eq. (21) from the Wasserstein distance, and mean, which disregards the information regarding the variance). For the moment, we report this for two ResNet architectures and one MLP, all trained with CIFAR10. As we can see, we consistently get better results when we use the proper full objective. For the next version of the manuscript, we will make sure to complete the table.
>
> | Method              | Model      | Barrier      |
> | ------------------- |:---------- | ------------ |
> | Aligned (full)      | ResNet20x2 | **1.486** |
> | Aligned (mean only) | ResNet20x2 | 1.492     |
> | No alignment        | ResNet20x2 | 1.911     |
> | Aligned (full)      | ResNet20   | **1.789** |
> | Aligned (mean only) | ResNet20   | 1.993     |
> | No alignment        | ResNet20   | 2.067     |
> | Aligned (full)      | MLP        | **0.028** |
> | Aligned (mean only) | MLP        | **0.028** |
> | No alignment        | MLP        | 0.880     |

---

> > ### Comment · Reviewer_Kqgj · 2023-08-17
> > **Response to the rebuttal**
> >
> > Thank you for running additional experiments. It does appear that  adding covariance to the objective indeed leads to relatively smaller loss barrier. I will keep my original score.

---

### Author Rebuttal · Authors · 2023-08-08

# General comments

First, we would like to thank the Reviewers for their comments and helpful feedback.
With this paper we analyze the geometry of the Bayesian posterior in deep neural networks by considering the permutation symmetries raising from the neural network parameterization. We do it by extending previous analysis on loss-optimized models to the Bayesian setting: this requires, among other things, to extend the concept of barrier and solutions interpolation.
In the variational inference setting, we move then to propose a methodology to align distributions, by framing this problem as a combinatorial optimization one. With this setup, we can find distributions that once interpolated exhibit low/zero barriers even for ResNet-scale models.

We are delighted to see that Reviewers agree on the novelty and quality of presentation (bvHV, Kavz, Kqgj, T6Qt, 2rWU), that our proposal is principled (2rWU) with rigorous experiments (bvHV, 2rWU) and interesting results (2rWU, Kqgj).

We use this space to address some common points:

* **Figures presentation**: several Reviewers (Kavz, 2rWU, hffp) commented on the readability of some of the plots in the submission. Indeed we had to squeeze the figures more than what we would have liked as to remain in the 9 pages, while presenting all the important results. We plan to remake the figures to address this point, possibly by converting the plots to LaTeX using tikzplotlib. Additionally, if allowed one additional page for the camera ready, we will be less constrained in the positioning of the figures, which should help.

* **Scope of the work and extensions beyond variational inference**: Reviewers 2rWU and T6Qt highlights how our initial conjecture is too broad, as it can cover all possible approximations (VI, Laplace, MCMC) while our work primarily focuses on VI. The idea for such a conjecture is to encourage further investigations with other setups and approximations. Nonetheless, we generally agree with this opinion and we will re-frame the Conjecture to better align with our context.

* **Data augmentation (DA)**: Reviewers bvHV and T6Qt expressed a limitation in our choice of not using data augmentation for our experimental setup. This choice was merely a consequence of the difficult interpretation of DA in the Bayesian context (as many previous works in the literature have shown, [e.g. from the paper 92,68,40]). Indeed, without DA (and without BatchNorm for normalization layers), we are provably targeting a correct Bayesian posterior. Having said that, for a sanity check we have run a ResNet20 on CIFAR10 with and without data augmentation, showing similar behaviour in both cases (Figure 8 in the rebuttal PDF).

* **Mixtures vs OT interpolation**: Reviewers 2rWU and T6Qt requested additional clarification on the choice of using optimal transport for the interpolation. While we address this point individually for both Reviewers, we want to give a generic intuition for this choice. We are interested in studying the properties of the approximate Bayesian posterior while we align the solutions w.r.t. permutations symmetries. With mixtures we essentially re-weighting the two solutions, without continuous "transport" of distribution mass between the two extremes. This means that if we look at the barrier, we don't see any not because they don't exist, but simply because we are interpolating in such a way that prevents us from exploring the geometry of the posterior. Said differently, mixtures do not lead us to the construction of low barrier solutions from permutation symmetries.

Below, a summary for the additional experiments run during the rebuttal and plots that you will find in the rebuttal PDF:

* **Figure 1**: Visualization of mixtures vs OT interpolation
* **Figures 2/3**: Two new visualizations of the posterior for aligned and not aligned solutions
* **Figure 4**: Analysis of features variance collapse in interpolated models
* **Figure 5**: Replication of Figure 3 from the paper, but using accuracy rather than likelihood
* **Figure 6**: Zoom of Figure 3 from the paper, with focus on the aligned models only
* **Figure 7**: Analysis of prior variance and posterior temperature on the solution connectivity for ResNet20
* **Figure 8**: Analysis of the effect of data augmentation on ResNet20

Here are the changes planned for the next version/camera-ready of the paper:

1. Change the conjecture in Sec. 1 to contextualize better our contributions
2. Include arguments for using optimal transport to interpolate solutions in Sec. 3
3. Fix the readability of some plots
4. Fix various typos and add a couple of missing references pointed out by the Reviewers
5. Add and comment the new experiments presented in this rebuttal on prior and temperature using ResNet20 models in Sec. 5
6. Add and comment the new experiments presented in this rebuttal on data augmentation in Sec. 5

Furthermore, the remaining discussion point of this rebuttal will be added to the Appendix.

Finally, in the threads below we will address the Reviewers' comments and questions in detail.

---

### Decision · Program_Chairs · 2023-09-21

**Decision:**

Accept (poster)

**Comment:**

The paper proposes an informal conjecture of linear connectivity modulo permutations for Bayesian neural networks trained with variational inference. The authors present some experiments supporting the conjecture, as well as an algorithm for aligning a pair of Bayesian neural networks.
I expect the authors to implement the suggestions made by the reviewers around Conjecture 1. In addition to weakening the conjecture to be more specific to mean field variational Bayesian inference, I would also suggest changing the language around “functionally transparent permutations”, which is undefined jargon. While conjecture 1 is already informal, this jargon makes the conjecture hard to interpret.
The paper could be further improved by adding the discussion on the connection between SGD and VI (as in the response to Reviewer 2rWU).